# Covalent linkage of the DNA repair template to the CRISPR-Cas9 nuclease enhances homology-directed repair

Natasa Savic[1†], Femke CAS Ringnalda[1†], Helen Lindsay[2,3], Christian Berk[4], Katja Bargsten[5], Yizhou Li[4], Dario Neri[4], Mark D Robinson[2,3], Constance Ciaudo[1], Jonathan Hall[4], Martin Jinek[5], Gerald Schwank[1]*

[1]The Institute of Molecular Health Sciences, ETH Zurich, Zurich, Switzerland; [2]The Institute of Molecular Life Sciences, University of Zurich, Zurich, Switzerland; [3]SIB Swiss Institute of Bioinformatics, Zurich, Switzerland; [4]Institute for Pharmaceutical Sciences, ETH Zurich, Zurich, Switzerland; [5]Department of Biochemistry, University of Zurich, Zurich, Switzerland

**Abstract** The CRISPR-Cas9 targeted nuclease technology allows the insertion of genetic modifications with single base-pair precision. The preference of mammalian cells to repair Cas9-induced DNA double-strand breaks via error-prone end-joining pathways rather than via homology-directed repair mechanisms, however, leads to relatively low rates of precise editing from donor DNA. Here we show that spatial and temporal co-localization of the donor template and Cas9 via covalent linkage increases the correction rates up to 24-fold, and demonstrate that the effect is mainly caused by an increase of donor template concentration in the nucleus. Enhanced correction rates were observed in multiple cell types and on different genomic loci, suggesting that covalently linking the donor template to the Cas9 complex provides advantages for clinical applications where high-fidelity repair is desired.

DOI: https://doi.org/10.7554/eLife.33761.001

*For correspondence: schwankg@ethz.ch

[†]These authors contributed equally to this work

**Competing interests:** The authors declare that no competing interests exist.

## Introduction

The CRISPR-Cas9 system is a versatile genome-editing tool that enables the introduction of site-specific genetic modifications (*Jinek et al., 2012*). In its most widespread variant a programmable chimeric single guide RNA (sgRNA) directs the Cas9 nuclease to the genomic region of interest, where it generates a site-specific DNA double-strand break (DSB) (*Mali et al., 2013*). In mammalian cells the repair of DSBs by different end-joining (EJ) pathways, such as classical non-homologous end joining (c-NHEJ), alternative non-homologous end-joining (a-NHEJ), or single-strand annealing (SSA) often leads to the formation of insertions or deletions (indels) (*Shalem et al., 2014*; *Ceccaldi et al., 2016*). Alternatively, when a donor template is provided, mammalian cells can also resolve DSBs via homology-directed repair (HDR) mechanisms, such as the classical homologous recombination (HR) pathway (*Mao et al., 2008*) and the Fanconi Anemia (FA) repair pathway (*Richardson, 2017*). While the formation of indels allows the elimination of gene function, repair from an ectopic donor oligonucleotide (oligo) via HDR mechanisms enables the introduction of DNA modifications with single base pair precision (*van den Bosch et al., 2002*).

Therapeutic applications of CRISPR-Cas9 generally require the precise correction of pathogenic mutations using donor templates. However, DSBs introduced in mammalian cells are predominantly repaired by error-prone EJ pathways. As the resulting indels inhibit the CRISPR-Cas9 complex from retargeting the locus, error-prone repair indirectly competes with HDR, and therefore reduces the rates of precise correction from donor templates. Furthermore, if the targeted allele is a hypomorph

**eLife digest** Genome editing allows scientists to change an organism's genetic information by adding, replacing or removing sections of its DNA sequence. The CRISPR-Cas9 system is a genome-editing tool that has had a large impact on biological research in recent years, and also shows promise for the treatment of patients with genetic disorders.

The tool works as follows: a small piece of RNA (a close cousin to DNA) is used to guide an enzyme called the Cas9 endonuclease to the desired region of the genome. Then, like a pair of molecular scissors, the enzyme cuts the DNA, breaking both strands of its double helix. The cell naturally starts to repair the damaged DNA, and one way to do this is to use another similar piece of intact DNA as a template. Scientists can exploit this repair mechanism (known as homology-directed repair) by giving the cell extra DNA that carries their desired sequence change, with the hope that the cell will use it as a template and edit its own genome in precisely the same way. However, it turns out that mammalian cells rarely use the template DNA to repair the damage. Instead, mammals tend to fix double-stranded breaks in DNA by simply joining the broken ends together, a method that is prone to errors.

To overcome this specific issue, Savic, Ringnalda et al. tested the effect of physically linking the template DNA to the Cas9 enzyme, so that the DNA was already nearby when the enzyme made the cut. Experiments with human cells confirmed that this new approach increased the frequency of homology-directed repair up to 24-fold compared to leaving the enzyme and the template DNA separate. Improving the CRISPR-Cas9 system in this manner makes it more likely that genome editing may one day become a routine treatment for patients with genetic disorders. But first, more preclinical studies are needed to assess the safety of the CRISPR-Cas9 technology for gene editing in patients.

DOI: https://doi.org/10.7554/eLife.33761.002

with residual gene function, the generated indels can further worsen the clinical phenotype of the disease (*Chu et al., 2015*). In recent years, several attempts have therefore been made to enhance HDR-mediated correction of CRISPR-Cas9-induced DSBs from donor oligos. Based on the knowledge that HDR pathways are primarily active during the S/G2 phase of the cell cycle, cells have been synchronized prior to CRISPR-Cas9 delivery (*Lin et al., 2014a*), and Cas9 expression has been limited to the S/G2/M phase of the cell cycle (*Gutschner et al., 2016*; *Howden et al., 2016*). Other studies have increased HDR by chemically modulating the EJ and HDR pathways (*Chu et al., 2015*; *Maruyama et al., 2015*; *Yu et al., 2015*; *Song et al., 2016*), and by rationally designing DNA repair templates with optimal homology arm lengths (*Richardson et al., 2016*). In addition, it has been proposed that the availability of the DNA repair template might present a rate-limiting factor for

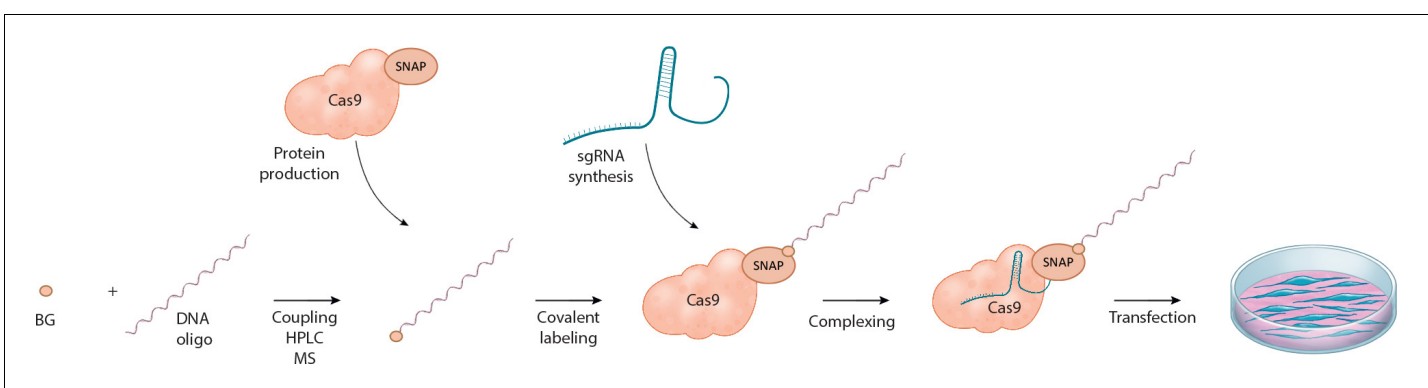

**Figure 1.** Schematic overview of the workflow for linking the DNA repair template to the Cas9 RNP complex. $O^6$-benzylguanine (BG)-labeled DNA oligos are covalently linked to Cas9-SNAP fusion proteins. The DNA-Cas9 molecules are then complexed with the specific sgRNAs to form the functional ribonucleoprotein-DNA (RNPD) complexes.

DOI: https://doi.org/10.7554/eLife.33761.003

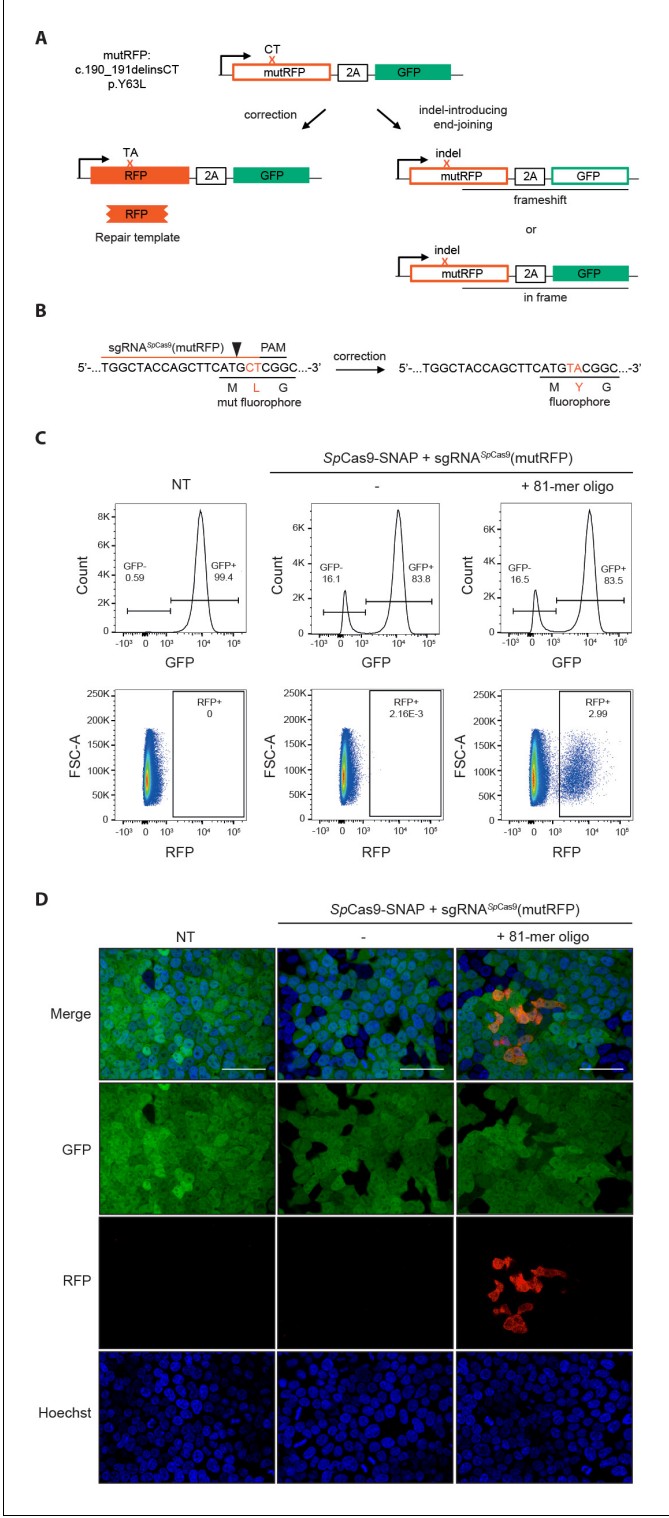

**Figure 2.** Fluorescent reporter system for high-throughput analysis of DSB repair rates. (a) Schematic overview of the HEK293T fluorescent reporter system. The RFP fluorophore carries a c.190_191delinsCT mutation that substitutes two nucleotides TA at the positions 190 and 191 in the RFP sequence to CT. This leads to the inactivation the RFP fluorophore by the substitution of tyrosine at the position 63 with leucine (p.Y63L). Repair of the mutation via donor oligos generates RFP/GFP double positive cells; indel mutations generate RFP/GFP double negative cells if they induce a frame shift. Of note, analysis of the reporter locus by next generation sequencing (NGS) demonstrated that 20 percent of indels did not lead to a frame shift (**Supplementary file 4**). Nevertheless,

*Figure 2 continued on next page*

*Figure 2 continued*

although FACS analysis thereby underestimates the absolute number of edited cells, it allows to accurately compare the correction efficiencies of different Cas9 systems. 'X' labels the mutation in RFP. 2A stands for 2A 'self-cleaving' peptide (*Kim et al., 2011*). (**b**) Schematic overview of the *Streptococcus pyogenes* sgRNA targeting mutRFP fluorophore and the corresponding PAM site. Black arrow indicates the introduced DSB site. The two nucleotides in the sgRNA seed sequence as well as the amino acid in the fluorophore region that are changed upon of precise repair (CT >TA and L > Y) are shown in orange. (**c**) Correction and indel rates can be quantified by FACS. The panels show FACS plots for gating GFP negative cells (upper panel) and the RFP positive cells (lower panel). (**d**) Representative confocal microscopy images. Scale bar: 50 µm, magnification 20x. Live cell nuclei were stained with Hoechst 33342. The efficiency of the sgRNA (sgRNA$^{SpCas9}$(mutRFP)) is shown in *Figure 3—figure supplement 1b*.

DOI: https://doi.org/10.7554/eLife.33761.004

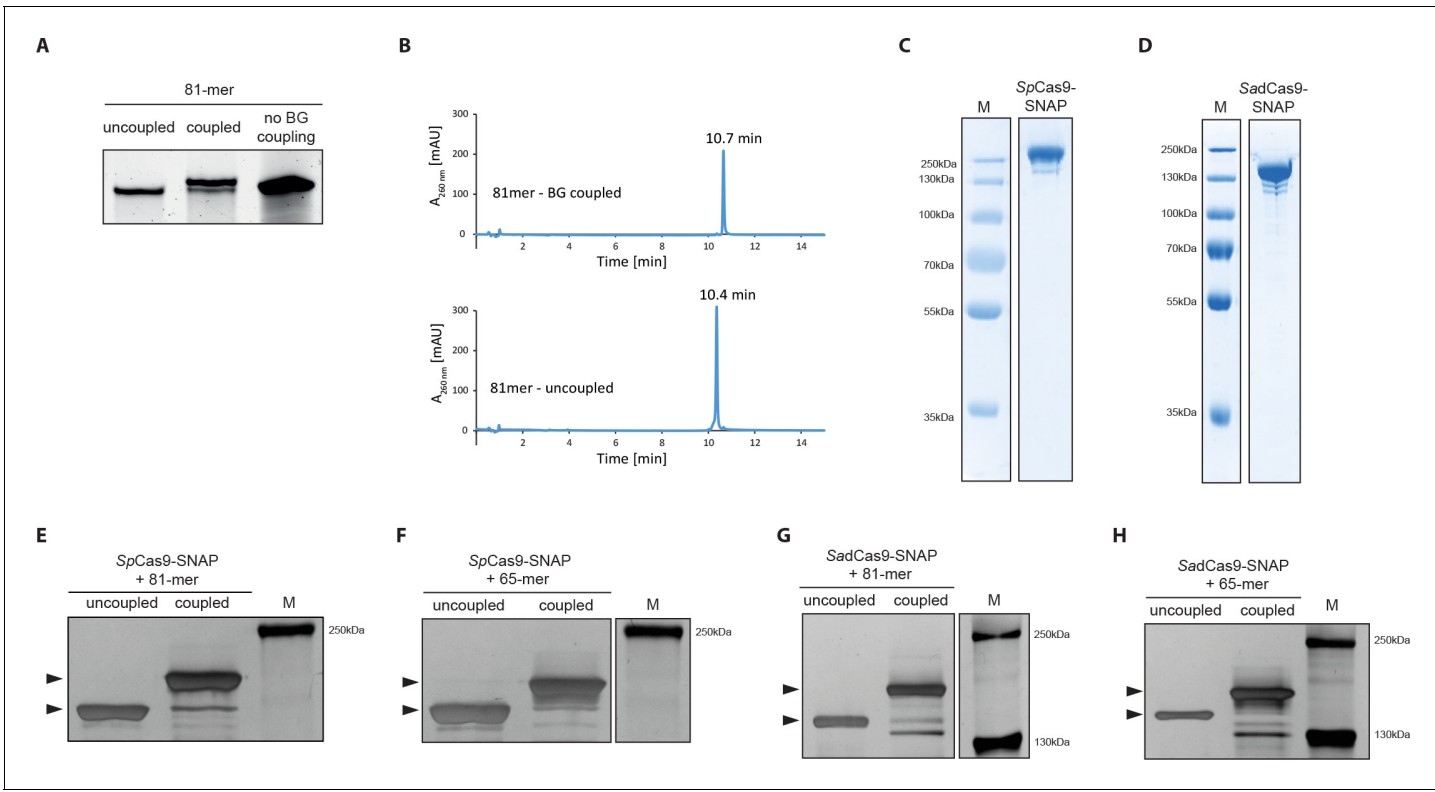

**Figure 3.** Covalent linkage of the DNA repair template to the Cas9 RNP complex. (**a**) Band shift of the 81-mer amino-modified oligo after coupling to BG-GLA-NHS shown on a denaturing PAGE gel. Amino modified oligos were mixed with amine-reactive BG building blocks and the samples were taken prior to the reaction (uncoupled) and after the reaction (coupled). No BG coupling: no amine-reactive BG building block was added to the amino modified oligos. (**b**) LC-MS analysis of HPLC-purified BG-coupled and uncoupled DNA repair templates. (**c,d**) Coomassie Blue stained SDS-PAGE gels of the purified *Sp*Cas9-SNAP and the *Sad*Cas9-SNAP fusion proteins (functionality of the SNAP-tags is shown in *Figure 3—figure supplement 1e,f*). (**e-h**) Silver stained SDS-PAGE gels. Band shifts confirm covalent linkage of Cas9-SNAP proteins to BG-coupled 81-mers. Lower arrowheads: unbound Cas9-SNAP. Upper arrowheads: Cas9-SNAP covalently bound to oligos.

DOI: https://doi.org/10.7554/eLife.33761.005

The following source data and figure supplement are available for figure 3:

**Source data 1.** Numerical data and the exact p values for all graphs in *Figure 3—figure supplement 1*.
DOI: https://doi.org/10.7554/eLife.33761.007

**Figure supplement 1.** Covalent linkage of the DNA repair template to the Cas9 RNP complex.
DOI: https://doi.org/10.7554/eLife.33761.006

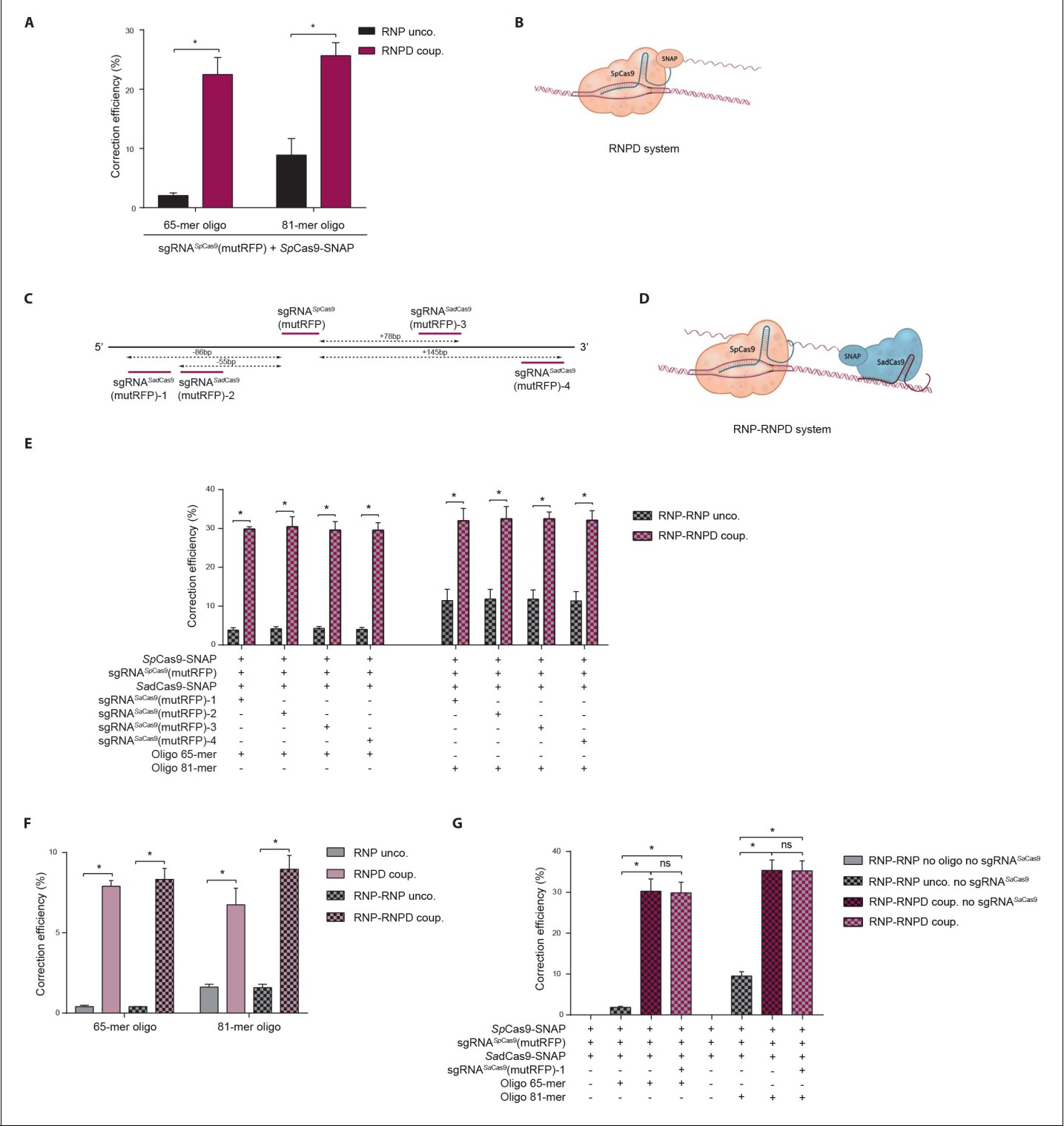

**Figure 4.** Linking the repair template to the Cas9 RNP complex enhances correction efficiency in a fluorescent reporter cell line. (a) Comparison between the control Cas9 system (RNP unco.: *Sp*Cas9-SNAP plus unlabeled donor oligo) and our novel system (RNPD coup.: *Sp*Cas9-SNAP conjugated to BG-labeled donor oligo). Cells were analyzed 5 days after transfection by FACS. Results are presented as correction efficiency (percentage of correction in edited cells). (b) Illustration of our novel Cas9 system, in which the repair template is covalently bound to *Sp*Cas9-SNAP (RNPD coup.). (c) Schematic overview of the binding positions of different *Sa*dCas9 sgRNAs (sgRNAs$^{SadCas9}$(mutRFP)-1-4) in comparison to the *Sp*Cas9 sgRNA targeting the mutRFP fluorophore (sgRNAs$^{SpCas9}$(mutRFP)). (d) Illustration of the two-component system, where the repair template is linked to the catalytically inactive *Sa*dCas9-SNAP (RNP-RNPD coup.). (e) Comparison between the two-component system (RNP-RNPD coup.: *Sp*Cas9-SNAP + *Sa*dCas9-

*Figure 4 continued on next page*

*Figure 4 continued*

SNAP bound to BG-labeled donor oligo) and the corresponding control Cas9 system (RNP-RNP unco.: *Sp*Cas9-SNAP + *Sa*dCas9-SNAP + unlabeled repair oligo). (f) Transfection of the one component system (grey and pink panels) and two component system (black/grey and black/pink panels) into reporter cells at a 5-time lower concentrations. In the two component system sgRNA$^{SadCas9}$(mutRFP)-3 was used. (g) Comparison of two component systems with and without the sgRNA for the *Sa*dCas9 complex. RNP unco. (*Sp*Cas9-SNAP + uncoupled oligo + sgRNA$^{SpCas9}$(mutRFP)); RNPD coup. (*Sp*Cas9-SNAP-coupled BG-oligo + sgRNA$^{SpCas9}$(mutRFP)); RNP-RNP unco. (*Sp*Cas9-SNAP + sgRNA$^{SpCas9}$(mutRFP) + *Sa*dCas9-SNAP + uncoupled oligo + sgRNA$^{SadCas9}$(mutRFP)); RNP-RNPD coup. (*Sp*Cas9-SNAP + sgRNA$^{SpCas9}$(mutRFP) + *Sa*dCas9-SNAP-coupled BG-oligo + sgRNA$^{SadCas9}$(mutRFP)). All values are shown as mean ±s.e.m of biological replicates; *$p<0.0332$ with n = 3 (f) and n = 4 (a,e,g) (n represents the number of biological replicates). A one-tailed Mann-Whitney test was used for comparisons. Numerical data and the exact p values for all graphs are shown in the *Figure 4—source data 1*.

DOI: https://doi.org/10.7554/eLife.33761.008

The following source data and figure supplements are available for figure 4:

**Source data 1.** Numerical data and the exact p values for all graphs in *Figure 4*.
DOI: https://doi.org/10.7554/eLife.33761.011

**Source data 2.** Numerical data for all graphs in *Figure 4—figure supplement 1*.
DOI: https://doi.org/10.7554/eLife.33761.012

**Source data 3.** Numerical data and the exact p values for all graphs in *Figure 4—figure supplement 2*.
DOI: https://doi.org/10.7554/eLife.33761.013

**Figure supplement 1.** Linking the repair template to the Cas9 RNP complex increases correction rates at the expense of indel formation.
DOI: https://doi.org/10.7554/eLife.33761.009

**Figure supplement 2.** Mechanistic insights into enhanced correction rates.
DOI: https://doi.org/10.7554/eLife.33761.010

HDR, and that enhancing the local concentration of donor oligos could increase the correction rates (*Ruff et al., 2014*; *Carlson-Stevermer et al., 2017*). Based on this hypothesis, we here generated and tested novel CRISPR-Cas9 variants, in which the DNA repair template is covalently conjugated to Cas9 (*Figure 1*).

## Results and discussion

To be able to measure HDR efficiencies of novel CRISPR-Cas9 variants in a rapid and high-through-put manner, we first generated a fluorescent reporter system (*Figure 2a–b*). In brief, the reporter cassette was stably integrated in HEK293T cells, and expresses a green fluorescent protein (GFP) that is preceded by an inactive mutant version of a red fluorescent protein (mutRFP). While precise correction of the mutation via HDR from donor templates leads to re-activation of RFP activity, the generation of frame shifts via error-prone EJ pathways leads to loss of GFP activity (*Figure 2a–b*). The correction and indel formation events can be visualized by fluorescence imaging and quantified by FACS (*Figure 2c–d*). To test the functionality of the reporter system, and to determine the optimal length of DNA repair templates, we first transfected Cas9-sgRNA ribonucleoprotein (RNP) complexes and single stranded (ss) oligo repair templates of different lengths (*Figure 3—figure supplement 1a*). In line with previous studies (*Zuo et al., 2017*), we found that maximal DSB correction rates are reached with ss-donor oligos of approximately 80 bases. We therefore continued our study with 81-nucleotide (81-mers) ss-donor oligos but also included 65-nucleotide (65-mers) oligos, as we reasoned that if repair templates are brought in proximity to DSBs also shorter homology arms could be sufficient for HDR.

In order to link the donor oligos to the Cas9 protein, we used the SNAP-tag technology, which allows covalent binding of $O^6$-benzylguanine (BG)-labeled molecules to SNAP-tag fusion proteins (*Keppler et al., 2003*). To generate $O^6$-benzylguanine (BG)-linked DNA repair templates, we first coupled amine-modified oligos to commercially available amine-reactive BG building blocks (*Figure 3a*, *Figure 3—figure supplement 1c*). The BG-linked oligos were further separated from unreacted oligos by HPLC (*Figure 3b*) and analyzed by liquid chromatography-mass spectrometry (LC-MS) to confirm their purity (*Figure 3—figure supplement 1d*). Next, we produced recombinant Cas9 proteins with a SNAP-tag fused to the C-terminus (*Figure 3c,d*). The fusion proteins were then complexed with the BG-coupled oligos, and covalent binding was confirmed by SDS-PAGE (*Figure 3e–h*). The protein-oligo conjugate was mixed with in vitro transcribed sgRNAs targeting the

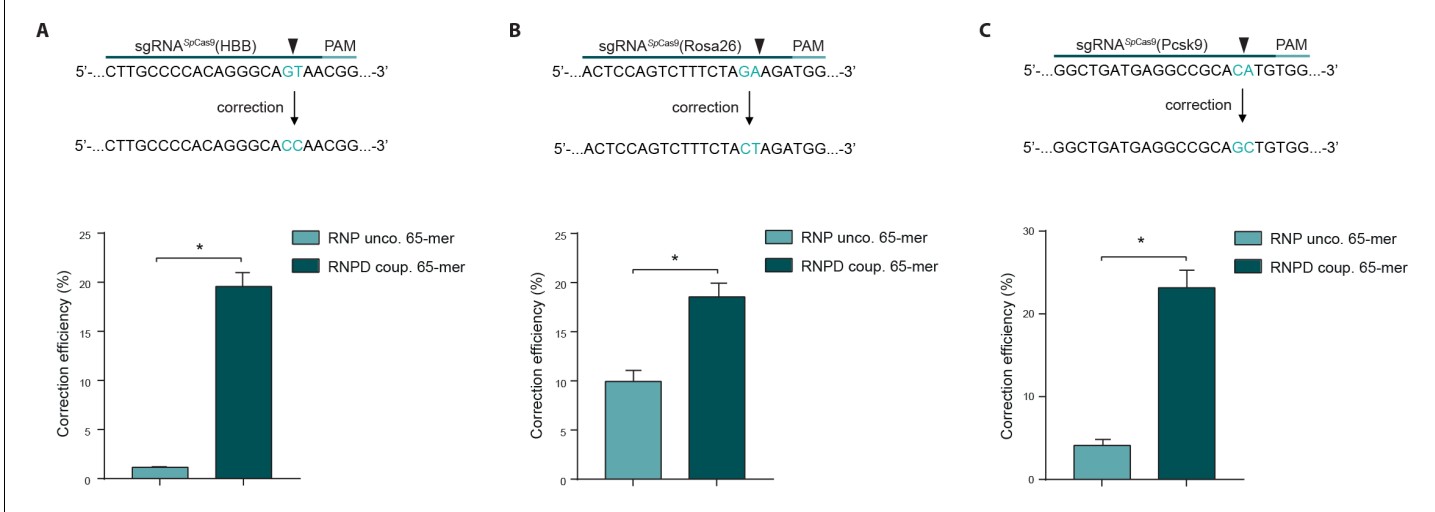

**Figure 5.** Linking the repair template to the Cas9 RNP complex enhances correction efficiency at endogenous loci. (**a,b,c**) Upper panels: Schematic overview of the target genomic regions of the *Streptococcus pyogenes* gRNAs. Black arrow indicates the introduced DSB site. The nucleotides that are exchanged in case of precise repair are shown in blue. Lower panels: NGS data quantification: Correction efficiency of the control Cas9 system (RNP unco.: *Sp*Cas9-SNAP plus unlabeled donor oligo) compared to our novel system (RNPD coup.: *Sp*Cas9-SNAP bound to BG-labeled donor oligo) is shown. (**a**) The HBB locus was targeted in a K562 cell line. The (**b**) Rosa26 and (**c**) Pcsk9 loci were targeted in mouse ESCs. All values are shown as mean ±s.e.m of biological replicates; *p<0.0332 with n = 4 (**a**) and n = 3 (**b,c**) (n represents the number of biological replicates). A one-tailed Mann-Whitney test was used for comparisons. Allele plots, variant count tables and categorized variant count tables for these loci are available as *Supplementary file 2–4*. Numerical data and the exact p values for all graphs are shown in the *Figure 5—source data 1*.
DOI: https://doi.org/10.7554/eLife.33761.014

The following source data is available for figure 5:

**Source data 1.** Numerical data and the exact p values for all graphs in *Figure 5*.
DOI: https://doi.org/10.7554/eLife.33761.015

mutRFP locus (*Figure 3—figure supplement 1g,h*), finally generating the Cas9 ribonucleoprotein-DNA (RNPD) complex.

To test if linking the donor oligo to Cas9 changes the ratio between indel formation and correction from the repair template, we used our reporter system to compare *Streptococcus pyogenes* Cas9 (*Sp*Cas9) complexes with linked repair oligos (*Figure 4b*) to the control *Sp*Cas9 complexes with unlinked repair oligos (*Sp*Cas9-SNAP with unlabeled oligos). Notably, the correction efficiency (percentage of corrections in edited cells) with bound complexes was significantly enhanced, from 2.1% to 22.5% with the 65-mers and from 8.9% to 25.7% with the 81-mers (*Figure 4a*, *Figure 4—figure supplement 1a,b*). In comparison to unbound complexes this represented 11- and 3-fold increases, respectively.

To further test our hypothesis that spatial and temporal co-localization of repair templates and Cas9 enhances correction rates, we next developed an independent approach where the donor oligo is not bound to the Cas9 complex that induces the DSB, but to a second catalytically inactive Cas9 complex that binds in close proximity to the DSB (RNP-RNPD system). To avoid the interchange of sgRNAs between both complexes, we designed a two-component system in which the DSB is induced by *Sp*Cas9, and the repair template is linked to a catalytically inactive *Staphylococcus aureus* (*Sa*)dCas9 (*Figure 4c,d*). We co-transfected both complexes into the reporter cell line, and quantified correction and indel formation rates. Notably, the correction efficiency increased from 4.1% to 29.9% with 65-mers, and from 11.6% to 32.3% with 81-mers (*Figure 4e*, *Figure 4—figure supplement 1c,d*), confirming our previous results with the RNPD system.

In vivo, the delivery efficiency of RNPs and oligos is generally lower than in vitro. Thus, if the repair template is not bound to Cas9, there is a substantial probability that only one of the two components would be delivered into the cell. In addition, at lower transfection efficiencies fewer repair templates are present in the nucleus, potentially decreasing HDR rates. As we presumed that linking the donor oligo to Cas9 should largely alleviate these limitations, we investigated whether the repair

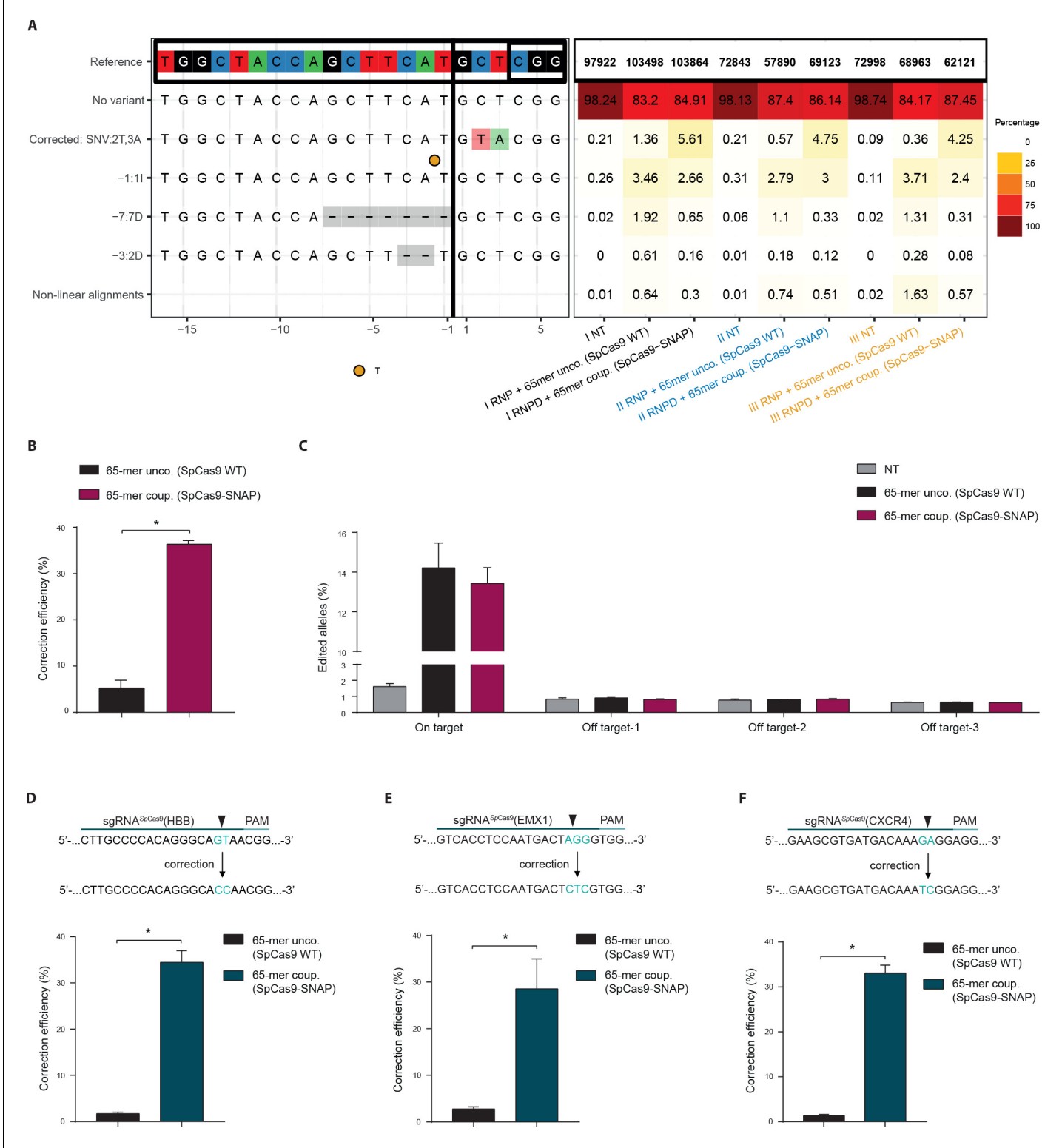

**Figure 6.** Direct comparison of the Cas9 RNPD system to the classical Cas9 complex. Classical Cas9 system (wild type *Sp*Cas9 plus unlabeled donor oligo); Our novel RNPD system (*Sp*Cas9-SNAP conjugated to BG-labeled donor oligo). (**a**) Targeting of the reporter locus in HEK293T cells. Illustration of the most frequent variants found by NGS in untreated samples (NT), in samples transfected with the classical Cas9 system (65-mer unco. (*Sp*Cas9 WT)), and in our engineered system (65-mer coup. (*Sp*Cas9-SNAP)). Alleles with a frequency above 0.5% in any of the nine samples are shown. For alleles with lower frequencies see *Supplementary file 3*. Abbreviations: Insertion (I), Deletion (D), Single nucleotide variant (SNV). Different colours in the x-axis indicate the three experimental replicates. A detailed description of the plot labels can be found in the Supplementary File 2 legend. (**b,d,e,f**)

*Figure 6 continued on next page*

Figure 6 continued

NGS data quantification of the (**b**) reporter locus, (**d**) HBB locus, (**e**) EMX1, and (**f**) CXCR4 locus targeted in HEK293T cells. In (**a,b,c**) the mutRFP sgRNA (see **Figure 2b**) was used. (**c**) Off target analysis for sgRNA$^{SpCas9}$(mutRFP): the percentage of edited alleles detected using NGS in untreated samples, in samples transfected with the classical Cas9 system, and in our engineered system. Information on the off target loci can be found in **Supplementary file 1** – Supplementary Table 1. (**d,e,f**) Upper panels: Schematic overview of the target genomic regions of the gRNAs. Black arrow indicates the introduced DSB site. The nucleotides that are exchanged in case of precise repair are shown in blue. Lower panels: NGS data quantification. Correction efficiency of the classical Cas9 system compared to our novel system is shown. All values are shown as mean ±s.e.m of biological replicates. *p<0.0332 with n = 3 (**b,c**) and n = 4 (**d,e,f**) (n represents the number of biological replicates). A one-tailed Mann-Whitney test was used for comparisons. Allele plots, complete variant count tables and categorized variant count tables for these loci are available as **Supplementary file 2–4**. Numerical data and the exact p values for all graphs are shown in the **Figure 6—source data 1**.

DOI: https://doi.org/10.7554/eLife.33761.016

The following source data is available for figure 6:

**Source data 1.** Numerical data and the exact p values for all graphs in **Figure 6**.

DOI: https://doi.org/10.7554/eLife.33761.017

efficiency with template-conjugated Cas9 is affected when complexes are transfected at 5-fold lower concentrations. Importantly, although under these conditions the correction efficiencies were generally lower with both coupled and uncoupled Cas9 complexes, the difference between the two systems was however even more pronounced. Compared to the uncoupled RNP complex, the RNPD system yielded 20-fold and a 4-fold increases in repair efficiency with 65-mer and 81-mer repair template oligos, respectively (**Figure 4f**, **Figure 4—figure supplement 1e,f**). Similarly, the two-component RNP-RNPD system led to a 21-fold increase with 65-mers and a 6-fold increase with 81-mers (**Figure 4f**, **Figure 4—figure supplement 1e,f**). Taken together, our results suggest that linking the repair template to the Cas9 complex leads to improved correction efficiency compared to the unlinked control CRISPR-Cas9 system, and that this effect is even more pronounced when CRISPR-Cas9 components are delivered at lower concentrations.

Next, we aimed to gain mechanistic insight into the processes that lead to enhanced correction rates when the donor oligo is linked to the Cas9 complex. We first assessed if the BG-labelling of the donor oligo itself already influences the correction efficiencies, and compared the correction rates of wild-type *Sp*Cas9 lacking the SNAP-tag together with either unlabelled oligo or BG-labelled oligo. While the correction rates were enhanced when the donor oligo was labelled with BG, the increase was several fold lower compared to the system where the oligo was conjugated to the Cas9 RNP complex (**Figure 4—figure supplement 2a–c**). We then investigated if the observed improvement in correction efficiency is due to the donor oligo being brought in close proximity to the DSB, or if it is sufficient to direct the donor oligo to the nucleoplasm. We therefore again employed the two-component system with DSB inducing *Sp*Cas9 and catalytically inactive *Sad*Cas9 conjugated to the donor oligo. While in one group *Sad*Cas9 was complexed with a sgRNA that directs it in close proximity to the DSB, in the other group the sgRNA was omitted and *Sad*Cas9 was therefore only directed into the nucleus. Importantly, our results demonstrated that adding the sgRNA did not further enhance correction rates, suggesting that the increase of donor oligo concentration in the nucleoplasm was sufficient to fully account for the positive effect of the Cas9-donor oligo conjugation (**Figure 4g**, **Figure 4—figure supplement 2d,e**). In line with these observations, a number of previous studies demonstrated that exogenous DNA transport into the nucleus is one of the major barriers to effective gene delivery (**Subramanian et al., 1999**; **Zanta et al., 1999**; **Ludtke et al., 1999**; **Aronsohn and Hughes, 1998**).

To validate our results from the HEK293T reporter cells, we next tested our approach at different endogenous genomic loci and in different cell types. We first targeted the human beta globin (HBB) locus in the K562 cell line, and analyzed correction and editing frequencies using next generation sequencing (NGS). The mean correction efficiency with the RNPD system was 19.6%, which represented a 17-fold increase compared to the control RNP system (**Figure 5a**, **Supplementary file 2**). Next we targeted the Rosa26 and proprotein convertase subtilisin/kexin type 9 (Pcsk9) locus in mouse embryonic stem cells (mESCs). Again, the mean correction efficiencies of RNPD systems were significantly increased, to 18.6% at the Rosa26 locus and 23.2% at the Pcsk9 locus (**Figure 5b,c**, **Supplementary file 2**). In comparison to the uncoupled RNP complexes, this represented 2- and 6-fold increases, respectively.

In the previous experiments the RNPD system was always compared to Cas9 SNAP-tag fusion proteins with uncoupled donor oligos. To also directly compare the engineered RNPD system to the classical CRISPR-Cas9 system, we performed experiments where we used wild-type Cas9 with the uncoupled donor oligos as a control. We first targeted the fluorescent reporter locus and analyzed it by NGS. We found that while the mean percentage of corrected loci increased from 0.8% with the classical Cas9 system to 4.9% with the RNPD system, the number of incorrectly edited loci slightly decreased from 12.6% to 9.3% (*Figure 6a*, *Supplementary file 2,3,4*). This corresponds to a 7-fold increase in correction efficiency (*Figure 6b*). In addition, the analysis of three computationally predicted off-target sites (*Lin et al., 2014b*; *Cradick et al., 2014*) of the reporter locus, suggests that the risk for generating off-target mutations is not enhanced with the RNPD system (*Figure 6c*, *Supplementary file 2,3,4*). In the next step we also targeted and analyzed the endogenous loci HBB, empty spiracles homeobox 1 (EMX1), and C-X-C chemokine receptor type 4 (CXCR4) in HEK293T cells. NGS analysis revealed that in all three loci the mean correction efficiency of the RNPD system was markedly increased to: 34,4% at the HBB locus, 28.6% at the EMX1 locus and 33.1% at the CXCR4 locus (*Figure 6d,e,f*, *Supplementary file 2,3,4*). Compared to the classical CRISPR-Cas9 system this represents a 20-fold, a 10-fold, and a 24-fold increase, respectively (*Figure 6d,e,f*).

Direct delivery of Cas9 RNP complexes into tissues promises great potential for therapeutic applications. Compared to genetically encoded systems, RNPs avoid the danger of genomic integration, and due to their limited lifetime, the risk of off-target activities is low (*Kim et al., 2014*). In addition, procedures for large-scale production of recombinant proteins for clinical use are well established, and several recently developed protocols enable in vivo delivery of Cas9 RNP complexes in animal models (*Wang et al., 2016*; *Zuris et al., 2015*; *Staahl et al., 2017*; *Lee et al., 2017*). Here, we present a method where we enhance correction efficiency of Cas9-induced DSBs by conjugating the donor oligo to the Cas9 complex. Our data suggests that the increase in HDR efficiency is caused by enhanced nuclear concentration of the repair template. Unlike previous approaches that increase HDR rates by chemically modulating DNA repair pathways, our approach does not alter endogenous cellular processes, thus reducing risk of potential negative side effects. In addition, covalent linkage of the repair template to the Cas9 RNP complex also addresses another central challenge of in vivo gene editing therapies – namely that simultaneous delivery of the RNP complex and the repair template needs to be ensured. Taken together, we suggest that covalently linking the DNA repair template to the Cas9 RNP complex is poised to further drive the CRISPR/Cas technology towards clinical translation.

## Materials and methods

Generation of CRISPR-Cas9 complexes with covalently bound repair templates is described in more detail at Bio-protocol (*Savić et al., 2019*).

**Key resources table**

| Reagent type (species) or resource | Designation | Source or reference | Identifiers | Additional information |
|---|---|---|---|---|
| Recombinant protein (*Streptococcus pyogenes*) | *Sp*Cas9-SNAP | This paper | | Schwank and Jinek lab |
| Recombinant protein (*Staphylococcus aureus*) | *Sad*Cas9-SNAP | This paper | | Schwank and Jinek lab |
| Genetic reagent | NH2-modified oligo | Integrated DNA Technologies | - | Custom DNA oligos/ '5 C6 NH2 modif. |
| Chemical compound | BG-GLA-NHS | New England Biolabs | ID_NEB:S9151S | |

Please see *Supplementary file 1*-Supplementary Tables 1–6 for a list of the DNA sequences used in this manuscript.

### Plasmids

All plasmids used in this study (listed in *Supplementary file 1*-Supplementary Table 6) have been deposited for the TULIPs system, along with maps and sequences, in Addgene.

Cloning of pNS19-LV-mutRFP-2A-GFP: pEGIP (addgene plasmid #26777) was mutagenized using QuikChange Lightning Multi Site-Directed Mutagenesis Kit (Agilent Technologies) to destroy the start codon of eGFP. Next the vector was linearized with BamHI and In-Fusion HD Cloning Plus CE (Takara) was used to insert the mutRFP-2A gBlocks Gene Fragment (Integrated DNA Technologies).

Cloning of pNS20-SpCas9-SNAP: pMJ922-SpyCas9-GFP bacterial expression vector was a kind gift from Prof. Martin Jinek. GFP was digested using BamHI and KpnI, and SNAPtag-NLS gBlocks (Integrated DNA Technologies) were integrated using In-Fusion HD Cloning Plus CE (Takara).

Cloning of pNS38-SadCas9-SNAP: pAD-SaCas9-GFP was generated by replacing the SpCas9 coding sequence in pMJ922 with SaCas9 sequence using Gibson cloning (Keppler et al., 2003). QuikChange Lightning Multi Site-Directed Mutagenesis Kit (Agilent Technologies) was used to remove the stop codon and to introduce the D10A and N580A mutations into the SaCas9 (SadCas9) gene. Subsequently, GFP was cut out using BamHI and KpnI, and replaced by a SNAP-tag-NLS gBlock (Integrated DNA Technologies) using In-Fusion HD Cloning Plus CE (Takara).

Plasmid pMJ806 was a gift from Jennifer Doudna (Addgene plasmid # 39312) (Jinek et al., 2012).

## Benzylguanine coupling reaction

Synthetic oligonucleotides with a 5′-Amino Modifier C6 functional group (100 µM) (Integrated DNA Technologies) were incubated with benzylguanine-GLA-NHS (1 mM) (NEB) and Hepes pH8.5 (200mM) for 60 min at 30°C. Coupling reactions were performed in following ratios: 30:1, 60:1 and 100:1 BG-GLA-NHS: amino modified oligo. After the coupling reaction all oligos were purified by ethanol precipitation. Repair oligo sequences can be found in *Supplementary file 1*-Supplementary Table 4.

## Denaturating PAGE

The benzylguanine (BG) coupled reactions were run on 20% polyacrylamide TBE gel containing 8M urea at 200 V for 60 min. The gel was stained for 30 min in 1x TBE containing Sybr Gold (Invitrogen), and imaged with a UV transilluminator (Biorad).

## HPLC purification and LC-MS analysis of the repair oligos

Benzylguanine coupled oligos were purified on an Agilent 1200 series preparative HPLC fitted with a Waters XBridge Oligonucleotide BEH C18 column, 10 × 50 mm, 2.5 µm at 65°C using a gradient of 5–25% buffer B over 8 min, flow rate = 5 ml min-1. Buffer A was 0.1 M triethylammonium acetate, pH 8.0. Buffer B was methanol. Fractions were pooled, dried in a speedvac and dissolved in H2O. Analysis of the purified BG-oligonucleotide was conducted on an Agilent 1200/6130 LC-MS system fitted with a Waters Acquity UPLC OST C18 column (2.1 × 50 mm, 1.7 µm) at 65°C, with a gradient of 5–35% buffer B in 14 min with a flowrate of 0.3 mL min−1. Buffer A was aqueous hexafluoroiso-propanol (0.4 M) containing triethylamine (15 mM). Buffer B was methanol.

## Expression and purification of Cas9-SNAP

Snap-tagged *Streptococcus pyogenes* Cas9 (*Sp*Cas9-SNAP), *Staphylococcus aureus* dCas9 (Sad-Cas9-SNAP) and Wild Type *Streptococcus pyogenes* Cas9 (SpCas9 WT) proteins were expressed in *Escherichia coli* BL21 (DE3) Rosetta 2 (Novagen) fused to an N-terminal fusion protein containing a hexahistidine affinity tag, the maltose binding protein (MBP) polypeptide sequence, and the tobacco etch virus (TEV) protease cleavage site. The cells were lysed in 20 mM Tris pH 8.0, 500 mM NaCl, 5 mM Imidazole pH 8.0. Clarified lysate was applied to a 10 ml Ni-NTA (Qiagen) affinity chromatography column. The column was washed by increasing the imidazole concentration to 10 mM and bound protein was eluted in 20 mM Tris pH 8.0, 250 mM NaCl, 100 mM Imidazole pH 8.0. To remove the His$_6$-MBP affinity tag, the eluted protein was incubated overnight in the presence of TEV protease. The cleaved protein was further applied to a heparin column (HiTrap Heparin HP, GE Healthcare) and eluted with a linear gradient of 0.1–1.0 KCl. The eluted protein was further purified by size exclusion chromatography using a Superdex 200 16/600 (GE Healthcare) equilibrated in 20 mM HEPES pH 7.5, 500 mM KCl.

## Covalent binding of Cas9-SNAP protein and BG-coupled oligonucleotide

Repair oligo templates coupled to BG were incubated with Cas9-SNAP proteins on the same day when the transfection is performed. BG-coupled oligos (2.2 pmols) were mixed with either *Sp*Cas9-SNAP or *Sa*dCas9-SNAP (2.2 pmols) and incubated for 60 min at 30°C. The negative controls (wild-type Cas9 +BG oligo or Cas9-SNAP + unlabeled oligo) were treated in the same way.

## SDS-PAGE gels

For confirming successful labeling of the Cas9-SNAP proteins with the BG-coupled oligonucleotides, BG-coupled and uncoupled oligonucleotides were mixed with either *Sp*Cas9-SNAP, *Sa*dCas9-SNAP or only the Cas9-SNAP proteins alone, reactions were incubated for one hour at 30°C. For the SNAP-Vista Green (NEB) substrate, the protein was incubated for 30 min on 30°C in the dark. After incubation, reactions (300 ng) were loaded on 6% SDS-PAGE gel and run at 80V for 160 min. Gels that were containing BG-Vista Green (NEB, SNAP-Vista Green), were imaged prior to silver staining. The green fluorescence signal of the SNAP-tag was detected with a UV transilluminator (Biorad). Subsequently, silver staining was completed using the Pierce Silver Stain Kit (Thermo Scientific) according to manufacturer instructions, and imaged with a UV transilluminator (Biorad).

## Production of sgRNAs

sgRNAs were generated from DNA templates using the T7 RNA Polymerase (Roche) in vitro transcription (IVT) kit. In short, sgRNA specific primers that also contain the T7 sequence were annealed with a common reverse primer that contains the sequence of the sgRNA scaffold (final concentrations 10 µM). DNA was purified with the QIAquick purification (Qiagen) kit and eluted in DEPC-treated water. PCR products were run on agarose to estimate concentration and to confirm amplicon size. In vitro transcription was performed at 37°C overnight. For purification, DNase I was added to the sgRNAs and incubated for 15 min at 37°C, and subsequently ethanol precipitation was performed overnight at −20°C. The sgRNAs were then further purified using RNA Clean and Concentrators (Zymo Research). Before use, all sgRNAs were checked on denaturing 2% MOPS gels. Complete sequences for all sgRNA protospacers, IVT primers and crRNAs can be found in *Supplementary file 1*-Supplementary Table 1, 2 and 3, respectively.

## Lentivirus production

HEK293T were PEI transfected with following plasmids: pNS19-LV-mutRFP-2A-eGFP, Pax2 and VSV-G. After 12 hr, the supernatant was discarded and changed to DMEM plus 10% FBS. 24 and 72 hr post-transfection, the media was collected and filtered through 0.45 µm filter and centrifuged at 20 000 G for 2:00 hr at 4°C. The pellet was then resuspended in 1 ml of DMEM and stored at −80°C.

## Fluorescent reporter generation

HEK293T cells were transduced with a lentiviral vector carrying the fluorescent reporter construct. Serial virus dilutions were used to isolate clonal populations using Puromycine selection (2 µg/ml) for 2 weeks.

## Cell culture and reagents

HEK293T cells were obtained from ATCC and verified mycoplasma free (GATC Biotech). The HEK293T reporter line was maintained in DMEM with GlutaMax (Gibco). Media was supplemented with 10% FBS (Sigma), and 100 µg/mL Penicillin-Streptomycin (Gibco). K562 cells were obtained from Sigma Aldrich, verified mycoplasma free and were maintained in RPMI 1640 medium with GlutaMax. Additional the medium supplemented with 10% FBS, and 100 µg/mL Penicillin-Streptomycin. Cells were passaged three times per week. Cells were grown at 37°C in a humidified 5% $CO_2$ environment. WT E14 mESC line (ATCC CRL-1821) was cultured in Dulbecco's Modified Eagle Media (DMEM) (Sigma-Aldrich), containing 15% of fetal bovine serum (FBS; Life Technologies), 100 U/mL LIF (Millipore), 0.1 mM 2-ß-mercaptoethanol (Life Technologies) and 1% Penicillin/Streptomycin (Gibco), on 0.2% gelatin-coated support in absence of feeder cells. The culture medium was changed daily. Cells were grown at 37°C in 8% $CO_2$.

## Transfection reactions

HEK293T cells were seeded in 24-well plates at 120.000–140.000 cells per well, 1 day prior to transfection. K562 cells were 6 hr prior to transfection distributed in 24 well plates at a density of 220.000–240.000 cells per well. On the day of transfection, RNP and RNPD complexes (2.2 pmols) were complexed with sgRNA (3.88 pmols) in Opti-MEM (Invitrogen) and briefly vortexed, followed by adding 3 µl the Lipofectamine 2000 reagent (Invitrogen) with Opti-MEM. The resulting mixture was incubated for 15 min at room temperature to allow lipid particle formation. After 15 min of incubation at room temperature, the mixture was dropped slowly into the well. One day post-transfection, cells were transferred to an 10 cm dish. mESCs were transfected into 6-well plate using Lipofectamine 2000. Cells were plated 24 hr before transfection at a density of 20,000 cells/cm$^2$ per well and cultured in culture medium without streptomycin and penicillin. The medium was changed to mESC culture medium 8 hr after transfection. Cells were collected 48 hr post-transfection.

## Flow cytometry analysis

For flow cytometry analysis, HEK293T reporter cells were analysed 5 days after transfection. Cells were trypsinized with TrypLE Express Enzym (Gibco), and resuspended in FACS buffer containing PBS/1% FBS/1% EDTA. Sytox Red was added for the exclusion of dead cells. Data were acquired on a BD LSR Fortessa cell analyser (Becton-Dickinson) and were further analysed using FlowJo software (FlowJo 10.2). In all experiments, a minimum of 200.000 cells were analysed. Gating strategy: Forward versus side scatter (FSC-A vs SSC-A) gating was used to identify cells of interest. Doublets were excluded using the forward scatter height versus forward scatter area density plot (FSC-H vs. FSC-A). Live cells were gated based on Sytox-Red-negative staining. Live-gated cells were further used to quantify the percentage of eGFP negative and turboRFP positive populations. Correction efficiency (%) or (percentage of corrections in edited cells) was culculated as 100 * (eGFP/turboRFP double positive population / (eGFP/turboRFP double negative population + eGFP/turboRFP double positive population)).

## Next Generation Sequencing

Transfected cells were collected by trypsinisation and were washed with PBS. PBS was discarded and DNA extraction was preformed using DNeasy Blood and Tissue kit (Qiagen) following manufacturer's protocol. The PCR amplicons flanking the targeted site were generated using NEBNext High-Fidelity 2X PCR Master Mix (NEB), primers that were used are listed in *Supplementary file 1*-Supplementary Table 5. PCR cycling conditions used were as follows: 1 × 98℃ for 3 min; 27 × 95℃ for 15 s, 65℃ for 15 s, 72℃ for 30 s; 1 × 72℃ for 5 min. Annealing temperature was optimized for each primer set to ensure that a single amplicon was produced. PCR amplicons were purified by solid phase reversible immobilization (SPRI) bead cleanup using Agencourt AMPure XP reagent (#A63881, Beckmann-Coulter, Indianapolis, IN, USA), per the manufacturer's instructions. For the generation of the pooled sequencing libraries, the TruSeq (Illumina) Index Adaptor Sequences were added at the second amplification step. The resulting Illumina libraries with Index Adaptors were purified with AMPure XP reagent. Quality control for the final library was performed using the High sensitivity D1000 ScreenTape at Agilent 2200 TapeStation. The libraries were sequenced using an Illumina MiSeq sequencer (MiSeq Reagent Kit v2 (15M, 500 cycle kit) or MiSeq Reagent Micro Kit v2 (4M, 500 cycle kit), Illumina, San Diego, CA, USA). Sequences were received in the format of demultiplexed FASTQ files produced by Illumina's bcl2fastq software (v2.19.0). Reads were merged with Pear v0.9.8 (*Zhang et al., 2014*) and merged reads mapped to the amplicon sequences with BWA-MEM v0.7.13-r1126 (*Li, 2010*). Unmerged reads were discarded. Sequence analysis was performed in R using CrispRVariants v1.9.0 (*Lindsay et al., 2016*). Variants within the protospacer +PAM region were analysed. Reads that align linearly and do not match the guide sequence are considered 'Edited'. Percentage of edited alleles is calculated as 100* Edited reads/Total reads (excluding non-linear alignments). Correction efficiency is 100* Perfectly corrected/Edited reads. The Scripts for mapping sequencing data, counting mutations and generating plots are available at https://github.com/HLindsay/Savic_CRISPR_HDR (*Lindsay, 2018*; copy archived at https://github.com/elifesciences-publications/Savic_CRISPR_HDR) Fastq files have been uploaded to ArrayExpress (*Brazma et al., 2003*), the accession number is E-MTAB-6808.

## Fluorescence microscopy

HEK293T reporter cells were imaged 7 days after transfection. Transfected cells were grown on Poly-L-lysine coated 8-well glass chamber slides (Vitaris) to 80–90% confluence. Hoechst 33342 (Thermo Scientific, Pierce) was added in the cell culture media to a final concentration of 0.1 µg/ml, and cells were incubated for 10 min at 37°C, 5% $CO_2$, prior to the image session. Confocal imaging was performed using a Leica DMI8-CS (ScopeM) with a sCMOS camera (Hamamatsu Orca Flash 4.0). The laser unit for confocal acquisition (AOBS system) contains 458, 477, 488, 496, 514 nm (Argon laser), 405 nm, 561 nm, 633 nm. Images were acquired using Leica LAS X SP8 Version 1.0 software, through using a 20 × 0.75 NA HC PLAN APO CS2 objective. Imaging conditions and intensity scales were matched for images presented together. Images were analysed using the Leica LAS AF (Lite) software version 3.3. Confocal images were processed using ImageJ software (Version 1.51 n).

## Statistical analyses

Statistical analyses were conducted using Graphpad's Prism7 software. A Mann-Whitney T-test was conducted for two-sample analyses (P value style: 0.1234(ns), 0.0332(*), 0.0021(**)). All values are shown as mean ± s.e.m of biological replicates. The number of biological replicates for each experiment was detailed in the Figure Legends. Numerical data and the exact p values for all graphs have been included in the Source data files.

## Data availability

The data that support the findings of this study are available within the paper, Supplementary files, Source data and NGS Fastq files have been uploaded to ArrayExpress.

## Acknowledgements

This work has been funded by the Swiss National Science Foundation PMPDP3_171388 (to NS), 31003A_160230 (to GS), 31003A_149393 (to MJ) and by the Vallee Scholar Award from The Bert L and N Kuggie Vallee Foundation (to MJ). The FACS analysis were performed at the ETH Flow Cytometry Core Facility (E-FCCF). Cell imaging was performed at the Scientific Center for Optical and Electron Microscopy (ScopeM) of the ETH Zurich. The NGS was performed at the Functional Genomics Center Zurich (FGCZ) of the ETH Zurich an the University of Zurich and Genetic Diversity Centre (GDC) of the ETH Zurich. We want to thank J. Huotari for critical feedback on the manuscript.

## Additional information

### Funding

| Funder | Grant reference number | Author |
| --- | --- | --- |
| Schweizerischer Nationalfonds zur Förderung der Wissenschaftlichen Forschung | PMPDP3_171388 | Natasa Savic |
| Schweizerischer Nationalfonds zur Förderung der Wissenschaftlichen Forschung | 31003A_160230 | Gerald Schwank |
| Schweizerischer Nationalfonds zur Förderung der Wissenschaftlichen Forschung | 31003A_149393 | Martin Jinek |
| Vallee Foundation | | Martin Jinek |

The funders had no role in study design, data collection and interpretation, or the decision to submit the work for publication.

### Author contributions

Natasa Savic, Conceptualization, Data curation, Formal analysis, Supervision, Funding acquisition, Validation, Investigation, Visualization, Methodology, Writing—original draft, Project administration,

Writing—review and editing, Conceived and designed the study, performed the experiments, Wrote the paper; Femke CAS Ringnalda, Data curation, Formal analysis, Validation, Investigation, Visualization, Methodology, Writing—original draft, Writing—review and editing, Performed the experiments, Wrote the paper; Helen Lindsay, Data curation, Software, Formal analysis, Investigation, Visualization, Writing—review and editing; Christian Berk, Investigation, Writing—review and editing, Performed HPLC and MS analysis; Katja Bargsten, Investigation, Methodology, Writing—review and editing, Produced the recombinant proteins; Yizhou Li, Investigation, Performed HPLC and MS analysis; Dario Neri, Methodology, Provided technical advice; Mark D Robinson, Software, Supervision, Writing—review and editing; Constance Ciaudo, Investigation, Writing—review and editing; Jonathan Hall, Methodology, Writing—review and editing, Provided technical advice; Martin Jinek, Supervision, Methodology, Writing—review and editing, Provided technical advice; Gerald Schwank, Conceptualization, Supervision, Funding acquisition, Validation, Investigation, Visualization, Methodology, Writing—original draft, Project administration, Writing—review and editing, Conceived and designed the study, Supervised the experiments, Wrote the paper

## Author ORCIDs

Natasa Savic (iD) https://orcid.org/0000-0003-3110-5780
Femke CAS Ringnalda (iD) http://orcid.org/0000-0002-0684-4613
Constance Ciaudo (iD) http://orcid.org/0000-0002-0857-4506
Jonathan Hall (iD) http://orcid.org/0000-0003-4160-7135
Martin Jinek (iD) http://orcid.org/0000-0002-7601-210X
Gerald Schwank (iD) http://orcid.org/0000-0003-0767-2953

## Decision letter and Author response

Decision letter https://doi.org/10.7554/eLife.33761.031
Author response https://doi.org/10.7554/eLife.33761.032

## Additional files

### Supplementary files

• Supplementary file 1. This file contains Supplementary Tables 1-6 (referenced in the Materials and methods). Supplementary Table 1 contains the guide protospacer sequences used in this study. Supplementary Table 2 contains the primer sequences for IVT of guides used in this study. Supplementary Table 3 contains the crRNA sequences of guides used in this study. Supplementary Table 4 contains the repair oligo sequences used in this study. The nucleotide substitution introduced by precise correction using repair template is shown in lowercase. Supplementary Table 5 contains the NGS primers used in this study. The target specific part of the primer is shown in uppercase, and the Illumina adapter is shown in lowercase. Supplementary Table 6 contains the plasmids used in this study.
DOI: https://doi.org/10.7554/eLife.33761.018

• Supplementary file 2. This file contains the allele plots for the loci analyzed by NGS. Allele plots show insertion/deletion variant alleles with frequency of at least 0.01%, and non-indel variants with frequency of at least 0.05% in any sample. When more than 50 variants passed these criteria, the top 50 alleles according to their maximum frequency in any sample are shown. From top to bottom, the consensus sequences for variant alleles are displayed in the order: no variant, precisely corrected allele, insertions (I) and deletions (D), single nucleotide variants (SNVs) and non-linear alignments. SNVs are only shown for non-indel variants and appear in color. In the y-axis labels, nucleotide numbers indicate the distance to the cut site. Variants are labelled with respect to the leftmost base. For example −5:9D is a 9 base pair deletion starting 5 bases upstream of the cut site. SNV labels show the bases that differ between the non-indel reads and the reference. The most common inserted sequences with less than 20 base pairs are shown in full in the legend. For longer and less frequent insertions the length is indicated. In the heatmap at right, the header shows the number of merged read pairs with alignments spanning the guide sequence. The x-axis is coloured according to experimental replicate.
DOI: https://doi.org/10.7554/eLife.33761.019

• Supplementary file 3. This file contains complete variant count tables for the genomic loci analyzed by NGS. Variants are labeled as in *Supplementary file 2*.
DOI: https://doi.org/10.7554/eLife.33761.020

• Supplementary file 4. This file contains categorized variant count tables for the genomic loci analyzed by NGS. Reads were classified as 'indel' if any insertions or deletions were present in the guide region, as 'no variant' if they perfectly matched the guide reference, (for on-target loci) 'corrected' if the targeted bases were changed as expected, and 'mismatch' if any other nucleotide changes were present.
DOI: https://doi.org/10.7554/eLife.33761.021

• Transparent reporting form
DOI: https://doi.org/10.7554/eLife.33761.022

## Data availability

The data that support the findings of this study are available within the paper and its Supplementary files. Source data files have been provided for Figure 4, Figure 5, Figure 6, Figure 3-Figure Supplement 1, Figure 4-Figure Supplement 1 and Figure 4-Figure Supplement 2. Scripts for mapping sequencing data, counting mutations and generating plots are available at https://github.com/HLindsay/Savic_CRISPR_HDR (copy archived at https://github.com/elifesciences-publications/Savic_CRISPR_HDR). Fastq files have been uploaded to ArrayExpress, and accession number is E-MTAB-6808.

The following dataset was generated:

| Author(s) | Year | Dataset title | Dataset URL | Database and Identifier |
|---|---|---|---|---|
| Helen Lindsay, Natasa Savic | 2018 | Fastq file from Covalent linkage of the DNA repair template to the CRISPR-Cas9 nuclease enhances homology-directed repair | https://www.ebi.ac.uk/arrayexpress/experiments/E-MTAB-6808/ | EMBL-EBI Array Express, E-MTAB-6808 |

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
