## [Decision Letter]

[Editors’ note: this article was originally rejected after discussions between the reviewers, but the authors were invited to resubmit after an appeal against the decision.]

Thank you for submitting your work entitled "Covalent linkage of the DNA repair template to the CRISPR/Cas9 complex enhances homology-directed repair" for consideration by *eLife*. Your article has been evaluated by a Senior Editor and three reviewers, one of whom is a member of our Board of Reviewing Editors. The reviewers have opted to remain anonymous.

Our decision has been reached after consultation between the reviewers. Based on these discussions and the individual reviews below, we regret to inform you that your work will not be considered further for publication in *eLife*.

The paper by Savic et al., "Covalent linkage of the DNA repair template to the CRISPR/Cas9 complex enhances homology directed repair" reports a novel approach to improve the efficiency of HR relative to NHEJ upon Cas9 induced DSB. The strategy relies on the assumption that enhancing the spatial proximity or local concentration of DNA used as template for HR would increase the use of this pathway. This is achieved by generating covalently coupled Cas9-oligonucleotide. An increased efficiency of HR is observed which could be interesting for some applications if confirmed.

However all reviewers have identified several problems in the manuscript that could only be solved pending designing a whole new set of experiments. This additional work extends beyond a revision and the manuscript cannot therefore be accepted for publication in *eLife*.

The major issues raised by the reviewers are:

- The effect observed with the coupled substrate does not demonstrate that it is due to bringing the oligonucleotide in proximity to the DSB.

- The effect observed should be validated on several genomic target sites, first to extend the observation to a genomic site and second to show the reproducibility of the effect on different targets.

- The assay used does not allow distinguishing the NHEJ and HR pathways as claimed by the authors.

Reviewer #1:

This paper reports a novel approach to improve the efficiency of HR relative to NHEJ upon Cas9 induced DSB. The strategy relies on the assumption that enhancing the spatial proximity or local concentration of DNA used as template for HR would increase the use of this pathway. This is achieved by generating covalently coupled Cas9-oligonucleotide. The increased efficiency of HR observed could be interesting for some applications.

The experiments are well designed and presented. However several aspects require clarifications as indicated below. Although the increased efficiency of HR is convincing the interpretation is open to several alternatives (i.e. proximity of oligo or stability of oligo). Two additional experiments could clarify this important point.

1) Figure 2A. The figure should be understandable with help of the legend and several pieces of information are missing:

What is 2A ? What is c.190_191delinsCT? (Use a more generic term; specific construct name can be provided in Materials and methods).

I assume the “X” labels the mutation and/or the Cas9 DSB?

I assume the mutant RFP is a substitution?

One needs to see exactly where the substitution is, where the guide maps and where the DSB is introduced and whether the guide would still be able to induce Cas9 cleavage after HR (is the PAM mutated?). One reason is that depending on where the DSB is, more or less end processing will be needed for HR.

2) Figure 2B lower panel (IF): indicate the channels used. Most of the cells seem to be GFP negative in the central panel. How could this be as the FACS indicate 16%? Single channels and overlays should be provided.

What is the percentage of GFP positive among the RFP positive cells? (Close to 100% expected).

It should be noted that the% of edited cells is underestimated since in frame indels (one third ?) can generate a GFP positive cell.

3) Figure 3A. Explain the lane no BG coupling?

4) Figure 4. The same data should not be plotted twice in panel B and C: Panel C should be removed (same comment for Figure 4F and G).

The main question about the use of *Sad*Cas9 is to determine whether the difference with *Sp*Cas9 is statistically significant. The tests should be performed (Figure 4E, between RNPD coup and RNP-RNPD coup) and if not significant the conclusions should be revised.

A map showing the positions of the different *Sad*Cas9 guides used (1 to 4) should be shown.

5) Two experiments are required to describe the effect observed: In order to know if the effect observed is due to coupling the oligo to Cas9, a control should be performed with the oligo coupled to BG but not to Cas9. Clearly at least part of the effect observed could be due to stabilization of the oligo rather than proximity to DSB per say. To distinguish these possibilities the authors should test an oligo coupled to *Sad*Cas9 but without the corresponding guide.

I assume *S. aureus* was used such as to design a distinct guide specific for *S. aureus* and not bound by *S. pyogenes*, if so, this should be explained in the text.

6) Figure 4—figure supplement 1.

What is the interpretation for the decrease of percentage of edited cells with oligo coupled to *Sad*Cas9 ? (Figure 4—figure supplement 1D).

Legend of Figure 4—figure supplement 1C and D: RNP-RNP unco is grey but should be hatched grey box.

7) Two-tailed tests should be used not one-tailed since there is no reason to assume a priori that the coupled protein would be more efficient.

8) Subsection “Expression and purification of Cas9-SNAP”, missing word: "The was further purified…"

Reviewer #2:

The authors show that covalent linkage of the DNA repair template to the CRISPR/Cas9 complex enhances HDR efficiency. They conclude this based on the use of a traffic light HDR reporter system, transiently transfected into HEK293 cells.

Although this is interesting and aspects of the study are really clever (e.g. the use of the sa-dCAS9 linked template to prove that physical proximity is the key), the work strikes me as quite preliminary as the increases are only shown in this transient reporter system using only one target site.

Without more thorough testing of the improvement in a real experimental system, I can't recommend acceptance of this study in its current form at *eLife*. My recommendations to improve the impact of this publication would be to demonstrate real efficiency improvements using a targeted edit within a genomic target site. In this respect, the existing work using transient reporter system would be excellent preliminary data for a real proof-of-concept – namely the increased efficiency in a real experimental genome engineering project e.g. to create a point mutation.

In addition, the CRISPR field has been populated by numerous claims showing positive perturbations to shift the balance of repair pathway from NHEJ to HDR (alterations in experimental design, NHEJ inhibitors, etc.). Frequently these claims have been made on the basis of data from manipulation of a single target site. Reproducibility of these studies has proven to be low, as when applied by other labs at different target sites, the reported improvements are frequently not replicated. To avoid this occurring, I would recommend that the authors address several different genomic target sites and report the level of improvement in HDR seen in these various experimental settings.

Figure 4E presents data which suggests that the positioning of the sgRNA (and hence the DSB) away from the target nucleotide to be mutated (0 bp, 52 bp, 61 bp, 83 bp and 128bp) has little influence on the correction efficiency. This is significantly at odds with experimental data from genomic target sites, and should be discussed. In addition, I wonder whether this is a curious artefact of using transient plasmids and thus provides support for my recommendation of addressing a single copy genomic target site or sites to validate their approach in a real experimental setting.

The work is very similar to a recent bioRxiv report by Janet Rossant's group (https://doi.org/10.1101/204339) who demonstrated that covalent attachment of the repair template to the CRISPR/Cas9 complex enhanced HDR rates, reporting data from fluorescent cassette insertions at 5 genomic target sites. Although this paper hasn't been peer reviewed, it's tempting to speculate that the authors of this manuscript have submitted what is essentially a preliminary study to *eLife* to compete with this more thorough study, which is presumably in review at another journal.

Reviewer #3:

Gene targeting constitutes a promising approach for the generation of novel biological models and for future gene therapy strategies. The capacity to generate targeted cleavage by the CRISPR-CAS9 system raised many hopes. However, targeted gene replacement still remains to too low levels. In the present work the authors design a strategy aiming at delivering the correcting DNA in the vicinity of the cleaved site. They covalently bound the correction oligonucleotide to the CAS9 itself via "click chemistry", using the SNAP-tag technology. Then they assemble in vitro the complex with the RNA-guide and transfected cells (HEK293). This strategy is elegant and promising. However, many concerns should be addressed.

The main problem is the fact that the authors used only on system to monitor HR; and that the designing of this substrate is based on wrong considerations on HR and NHEJ.

It is claimed that NHEJ is error-prone. This is wrong. There are two kinds of end-joining processes, the canonical NHEJ (which is not error-prone, but conservative) and the Alternative end-joining (alt-NHEJ, MMEJ, B-NHEJ), which is mutagenic and error-prone. Therefore mutagenic repair does not automatically imply NHEJ. Second it is said in the Introduction that HR and NHEJ directly compete. This is also wrong; in fact things happen in two phases: first competition between cNHEJ and single-strand DNA resection, second on resected DNA extremities, competition between HR and alternative end-joining. Finally, HR can also generate mutagenesis. Therefore many concerns exist on the strategy used here (which is the sole assay used) because mutagenic repair can arise by many other processes than NHEJ.

Moreover, is correction with oligonucleotides (65, 81 b) an actual HR mechanism? This is not consistent with concept of MEPS.

The authors should first genetically validate their reporter system, in cells mutated for HR or NHEJ.

The authors should also verify their strategy with natural endogenous target sequences, instead of the reporter.

There are no data on the transfection efficiency. Especially, comparing the CAS9 with the engineered one.

Similarly, does the two CAS9 cleave with similar efficiency?

Does the modification of the CAS9 affect its cleavage specificity?

Are there any off-target effects? (off-target cleavage, off-target integration of the oligo).

The authors should test different RNA guides for a common target.

The author should test different cell lines.

It is not clear what is actually measured. Where are the mutagenic repair (pseudo-NHEJ) measurements? Is it the frequency of HDR or the ratio HDR/pseudo-NHEJ?

[Editors’ note: what now follows is the decision letter after the authors submitted for further consideration.]

Thank you for resubmitting your work entitled "Covalent linkage of the DNA repair template to the CRISPR-Cas9 nuclease enhances homology-directed repair" for further consideration at *eLife*. Your revised article has been favorably evaluated by Diethard Tautz (Senior Editor) and the Reviewing Editor.

The manuscript has been improved but there are some remaining issues that need to be addressed before acceptance, as outlined below:

The authors provide convincing answers to all reviewers' comments. Importantly, the authors show that the main improvement achieved by their experimental approach is due to enhanced targeting of the oligonucleotides to the nucleus rather that its targeting to the specific target genomic site. This could also be highlighted in the Abstract.

A few points need to be clarified:

1) Source data 6 refers to Figure 4—figure supplement 2 (not 3?) should be registered as Figure 4—source data 3.

2) Source data 5 refers to Figure 4—figure supplement 1 (not 2?) should be registered as Figure 4—source data 2.

3) Figure 2B requires clarification (and in the legend as well). Legend says "the mutation substitutes… CT.. to TA". This is ambiguous because TA is wild type. It would clarify to draw in Figure 2, the wild type and mutated genomic sequences (with the codon substitution) and the sequence of the guide (also can be confusing as drawn because the guide has CT not TA).

4) Figure 5: In Figure 5, only the reference is shown. Mutated variant should be indicated. For instance, on 5A, at HBB, the DNA sequence shown should be the one after correction (to be consistent with Figure.2B) ? However it is identical to guide sequence?

Please explain the correction efficiency values, obtained after correction for transfection efficiency? If so, provide transfection efficiencies, otherwise read data from Supplementary file 2 cannot be understood: i.e. 27% at Rosa 26, how was this obtained? In addition, since no HR reads is detected at Rosa26 (why is this?), the authors should comment in the main text on the difference uncoupled/coupled from Figure 5B for this locus.

5) Figure 6A requires clarifications both in the main text and in the presentation of the figure. The authors summarize in one sentence all the NGS data (Results and Discussion, eighth paragraph). There is a lot of data in this analysis and it would be better to present the analysis step by step and not to refer to all panels 6A, B, D, E, F at once. Reporter locus could be presented first, with comments about percentage of corrected reads and other events (legend should explain the nomenclature, for instance -7:7D, and also what are I, II and III: triplicates I assume). Then results at other loci should be briefly discussed (and referred to Supplementary file 2). Also indicate in Supplementary file 2 that the variants called "SNV" are the ones predicted after repair by HR. An interesting information is also the relative proportion of HR versus non HR events. In Figure 6, it seems that this proportion is about 30% in experiment I (5.63/(100-85.19)). Is this correct? It could be discussed in the main text.

---

## [Author Response]

[Editors’ note: the author responses to the first round of peer review follow.]

The major issues raised by the reviewers are:- The effect observed with the coupled substrate does not demonstrate that it is due to bringing the oligonucleotide in proximity to the DSB.

To gain mechanistic insight into the process that leads to an increased correction rate when the donor oligo is coupled to the Cas9 complex, we performed the experiments reviewer 1 suggested. First, we tested if the increased repair rate is caused by BG-labelling of the donor oligo, and compared the editing rates of *Sp*Cas9 WT together with either unlabelled oligo or BG-labelled oligo (note that *Sp*Cas9 WT cannot bind BG-oligos). Although we observed a higher repair rate with the BG-labelled donor oligo, the increase was several fold lower compared to the experiment where the oligo was linked to the Cas9 complex (Figure 4—figure supplement 2A-C). Next, we tested if the observed improvement in correction efficiency is due to the donor oligo being brought in close proximity to the DSB, or if it is sufficient to direct the donor oligo to the nucleus. We therefore used our 2-component system (*Sad*Cas9 binds the oligo and *Sp*Cas9 generates the DSB) with and without the *Sad*Cas9 guide RNA. Our results demonstrate that targeting the donor oligo to the nucleus via *Sad*Cas9 fully explains the observed increase in HDR rates (Figure 4G, Figure 4—figure supplement 2D, E). In line with this hypothesis, previous studies have demonstrated that promoting active transport of plasmid DNA to the nucleus (e.g. by binding plasmids to proteins with a nuclear localisation signal (NLS)) enhances expression of genes located on these plasmids (doi: 10.1517/17425247.1.1.127).

- The effect observed should be validated on several genomic target sites, first to extend the observation to a genomic site and second to show the reproducibility of the effect on different targets.

We fully agree with the reviewers that confirming the results from the initial manuscript by targeting endogenous genomic loci in different cell lines/types is required to proof reproducibility and robustness of the system. For the revised manuscript we therefore targeted five endogenous loci in three different cell types (HEK cells, K562 cells, mESCs). Importantly, next generation sequencing revealed that linking the donor oligo to Cas9 robustly enhanced correction rates in all tested loci and cell types (Figure 5A-C, Figure 6D-F).

- The assay used does not allow distinguishing the NHEJ and HR pathways as claimed by the authors.

For the revised manuscript we analysed the sequence of the targeted HEK reporter locus as well as other endogenous loci by next generation sequencing (Figure 5A-C, Figure 6A-F). These data allowed us to directly measure the events in which the DNA break was precisely repaired from the donor oligo (e.g. in the reporter locus exchange of two base pairs at the DSB in the RFP fluorophore region), and events in which the DSBs led to indel mutations.

Nevertheless, as reviewer 3 correctly stated, we always measure DNA repair outcomes (corrections and indels) and not the activities of repair pathways. Since different pathways can trigger indel formation (e.g. c-NHEJ, A-NHEJ, or SSA) or correction from repair templates (e.g. HR or FA repair), it is indeed important to precisely define our wording. In the revised version of the manuscript we discuss this issue in the Introduction, and define homology-directed repair (HDR) as all mechanisms that can repair the locus from the donor oligo sequence, and end-joining (EJ) as all mechanisms that do not use the oligo template for DSB repair.

Of note, the main goal of our study was to develop a novel method for therapeutic gene editing, in which precise repair of DNA breaks from donor oligos is enhanced. Even though it would be very interesting to decipher which molecular pathways are involved in that process, we believe that this question would go beyond the scope of this study, and rather plan to address these topics in future investigations.

Reviewer #1:[…] The experiments are well designed and presented. However several aspects require clarifications as indicated below.Although the increased efficiency of HR is convincing the interpretation is open to several alternatives (i.e. proximity of oligo or stability of oligo). Two additional experiments could clarify this important point. In order to know if the effect observed is due to coupling the oligo to Cas9, a control should be performed with the oligo coupled to BG but not to Cas9. Clearly at least part of the effect observed could be due to stabilization of the oligo rather than proximity to DSB per say. To distinguish these possibilities the authors should test an oligo coupled to SadCas9 but without the corresponding guide.

Thank you for suggesting these elegant experiments. As explained above (major issue (i)) we performed both experiments, and found that targeting the donor oligo to the nucleus via *Sad*Cas9 fully explains the observed increase in HDR rates (Figure 4G, Figure 4—figure supplement 2D, E).

1) Figure 2A. The figure should be understandable with help of the legend and several pieces of information are missing:What is 2A?

2A stands for 2A “self-cleaving” peptide: a viral oligopeptide widely used for co-expression of multiple genes. The peptide employs ribosome skipping to mediate “cleavage” of polypeptides during translation in eukaryotic cells. In the revised manuscript we included this explanation in the figure legend.

What is c.190_191delinsCT? (Use a more generic term; specific construct name can be provided in Materials and methods).

According to the recommended mutation nomenclature c.190_191: denotes the mutation in the coding DNA in nucleotides in positions 190 and 191; delins stands for deletion / insertions (substitution of two nucleotides); TA: denote nucleotides that are present in the mutated sequence. In the revised version we clarified the term in the figure legend.

I assume the “X” labels the mutation and/or the Cas9 DSB?

“X” labels the mutation in RFP. In the revised manuscript we added an explanation to the figure legend.

I assume the mutant RFP is a substitution?

Yes, the mutated RFP has a substitution of two nucleotides in the fluorophore. In the revised manuscript we included a more detailed explanation in the figure legend.

One needs to see exactly where the substitution is, where the guide maps and where the DSB is introduced and whether the guide would still be able to induce Cas9 cleavage after HR (is the PAM mutated?). One reason is that depending on where the DSB is, more or less end processing will be needed for HR.

Thank you for this useful comment. In the revised version we included a map of the locus, which indicates the mutated turboRFP fluorophore sequence as well as the protospacer and PAM motif of the guide RNA (Figure 2B). In short, the mutated region is in the fluorophore domain of turboRFP, and upon correction the protospacer sequence is modified (2 bases at position 1 and 2). Thus, if repaired from the donor oligo, the locus cannot be retargeted.

2) Figure 2B lower panel (IF): indicate the channels used. Most of the cells seem to be GFP negative in the central panel. How could this be as the FACS indicate 16%? Single channels and overlays should be provided.

In the revised version we indicated the channels used for the IF (Figure 2D), and included the single channels next to the overlay images. The IF image is just a representation of how well one can distinguish the cells that have lost GFP and gained RFP in our reporter system. The percentage of the cells that lost GFP cannot be determined from one fluorescent image, since it contains only approximately 100 cells that are grown from a few clones. In FACS analysis we always quantify at least 200.000 cells, thus the data is more accurate, and as seen in our biological replicates very robust. In the revised version we show an IF image that is more representative to the FACS quantification.

What is the percentage of GFP positive among the RFP positive cells? (Close to 100% expected).

On average 99.7% of the cells that are RFP positive are also GFP positive (see Author response image 1).

It should be noted that the% of edited cells is underestimated since in frame indels (one third ?) can generate a GFP positive cell.

Thank you for this important comment. In the revised manuscript we illustrated in Figure 2A and described in the figure legend that the percentage of edited cells is underestimated when using FACS analysis as a readout. Importantly, the NGS experiments (Figure 6) allowed us to accurately quantify in-frame and out-of-frame indels. For the mutRFP reporter locus we found that on average 34% of indels are in-frame mutations. Although this leads to an underestimation of the absolute number of edited cells in our FACS reporter assay, it still allows to accurately compare the editing- and repair efficiencies of different Cas systems.

3) Figure 3A. Explain the lane no BG coupling?

Thank you for suggesting this clarification. The amino modified oligo BG coupling reaction was controlled in two ways:

1) uncoupled: where amino modified oligonucleotide was mixed with amine-reactive BG building block, but the sample was taken before the reaction has started.

2) no BG coupling: no amine-reactive BG building block was added to the sample.

In the revised version we included the explanation in the figure legend.

4) Figure 4. The same data should not be plotted twice in panel B and C: Panel C should be removed (same comment for Figure 4F and G).

Indeed, the same data were plotted twice; as percentage of the correction efficiency and as fold change. In the revised version we removed the plots for the fold change.

The main question about the use of SadCas9 is to determine whether the difference with SpCas9 is statistically significant. The tests should be performed (Figure 4E, between RNPD coup and RNP-RNPD coup) and if not significant the conclusions should be revised.

Thank you for the suggestion. We performed statistical analysis, and indeed we found that the differences were not significant in all groups (see Author response image 2). Since the main goal of using the two-component RNP-RNPD system was to show in a second and independent manner that the correction efficiency is enhanced when the donor oligo is locally concentrated, we decided to show the results from the two-component system without direct comparison to the one component system.

**Author response image 2. respfig2:** 

A map showing the positions of the different SadCas9 guides used (1 to 4) should be shown.

In the revised manuscript the map showing the positions of *Sad*Cas9 sgNAs have been added (Figure 4C, Supplementary file 1). The sequences and the positions of SaCas9 guides are given in respect to the turboRFP fluorophore mutation and the *Sp*Cas9 guide cut site.

5) Two experiments are required to describe the effect observed: In order to know if the effect observed is due to coupling the oligo to Cas9, a control should be performed with the oligo coupled to BG but not to Cas9. Clearly at least part of the effect observed could be due to stabilization of the oligo rather than proximity to DSB per say. To distinguish these possibilities the authors should test an oligo coupled to SadCas9 but without the corresponding guide.

Please see our first response to reviewer #1.

I assume S. aureus was used such as to design a distinct guide specific for S. aureus and not bound by S. pyogenes, if so, this should be explained in the text.

The reviewer is correct, the *S. aureus* Cas9 was used in combination with *S. pyogenes* Cas9 in order to make sure that guides are not interchanged between the Cas9 that is cleaving and the catalytically inactive Cas9 that holds the template. We added this explanation in the revised version of the manuscript.

6) Figure 4—figure supplement 1.What is the interpretation for the decrease of percentage of edited cells with oligo coupled to SadCas9 ? (Figure 4—figure supplement 1D).

Our hypothesis is that the constant presence of the donor oligo coupled to *Sad*Cas9 at the target locus could lead to interference of the ss-oligo with the DNA double-helix and disturb the cutting efficiency of the *Sp*Cas9.

Legend of Figure 4—figure supplement 1C and D: RNP-RNP unco is grey but should be hatched grey box.

Thank you for spotting this mistake. We corrected it in the revised manuscript.

7) Two-tailed tests should be used not one-tailed since there is no reason to assume a priori that the coupled protein would be more efficient.

We used the one-tailed t-test since we specifically want to test (the alternative hypothesis of) whether increasing the local oligo concentration at the DSB increased the correction efficiency (not whether the correction efficiency decreased). Notably, even when using the two-tailed t-test, the conclusions remain the same, namely that the difference in correction efficiency is statistically significant.

8) Subsection “Expression and purification of Cas9-SNAP”, missing word: "The was further purified…"

Thank you for pointing out this mistake. We corrected it in the revised manuscript.

Reviewer #2:[…] Although this is interesting and aspects of the study are really clever (e.g. the use of the sa-dCAS9 linked template to prove that physical proximity is the key), the work strikes me as quite preliminary as the increases are only shown in this transient reporter system using only one target site.

In the revised manuscript we described our reporter system in more detail to avoid any misunderstandings (Figure 2A). The fluorescent reporter system was not transiently transfected, but generated by a single stable integration of the reporter cassette into the genome of HEK293T cells.

Without more thorough testing of the improvement in a real experimental system, I can't recommend acceptance of this study in its current form at eLife. My recommendations to improve the impact of this publication would be to demonstrate real efficiency improvements using a targeted edit within a genomic target site. In this respect, the existing work using transient reporter system would be excellent preliminary data for a real proof-of-concept – namely the increased efficiency in a real experimental genome engineering project e.g. to create a point mutation.

As stated above, the reporter system was generated by a single stable integration into the genome of HEK293T cells. We nevertheless fully agree with reviewer 2 that analysis of different endogenous loci is necessary to validate our findings (see also our second response to major issues). In the revised manuscript we therefore tested our system in three different cell types on five different genomic loci (Figure 5A-C, Figure 6D-F). Importantly these data fully support our hypothesis that coupling the donor oligo to the Cas9 complex increases repair rates.

In addition, the CRISPR field has been populated by numerous claims showing positive perturbations to shift the balance of repair pathway from NHEJ to HDR (alterations in experimental design, NHEJ inhibitors, etc.). Frequently these claims have been made on the basis of data from manipulation of a single target site. Reproducibility of these studies has proven to be low, as when applied by other labs at different target sites, the reported improvements are frequently not replicated. To avoid this occurring, I would recommend that the authors address several different genomic target sites and report the level of improvement in HDR seen in these various experimental settings.

As mentioned above, in the revised manuscript we reproduced our findings in different cell types and endogenous loci by NGS (Figure 5A-C, Figure 6D-F).

Figure 4E presents data which suggests that the positioning of the sgRNA (and hence the DSB) away from the target nucleotide to be mutated (0 bp, 52 bp, 61 bp, 83 bp and 128bp) has little influence on the correction efficiency. This is significantly at odds with experimental data from genomic target sites, and should be discussed. In addition, I wonder whether this is a curious artefact of using transient plasmids and thus provides support for my recommendation of addressing a single copy genomic target site or sites to validate their approach in a real experimental setting.

Here has been a misunderstanding, and we therefore described the experiment in more detail in the revised version (Figure 4C). The DSB is always generated by the same *Sp*Cas9 sgRNA at position 0 from the mutation. 0 bp, 55 bp, 78 bp, 86 bp and 145bp indicates the distances of the different *Sad*Cas9 sgRNAs binding sites from the mutation in the RFP fluorophore (position 0).

The work is very similar to a recent bioRxiv report by Janet Rossant's group (https://doi.org/10.1101/204339) who demonstrated that covalent attachment of the repair template to the CRISPR/Cas9 complex enhanced HDR rates, reporting data from fluorescent cassette insertions at 5 genomic target sites. Although this paper hasn't been peer reviewed, it's tempting to speculate that the authors of this manuscript have submitted what is essentially a preliminary study to eLife to compete with this more thorough study, which is presumably in review at another journal.

In the mentioned report the authors use a Cas9-streptavidin fusion in combination with a biotinylated repair template to recruit the repair template to the editing site. This improves their targeting efficiencies. Since this is a non-covalent attachment system, it has certain disadvantages to the SNAP-tag system, and thus we consider our work complementary to their work. Of note, in our revised manuscript we also analysed several genomic loci in different cell types.

Reviewer #3:[…] The main problem is the fact that the authors used only on system to monitor HR; and that the designing of this substrate is based on wrong considerations on HR and NHEJ.

To clarify, using our reporter system we were able to detect Cas9-induced frame-shift indels (loss of eGFP fluorescence) and corrections from donor templates (activation of RFP). As explained in our third response to major issues, we fully agree with reviewer 3 that different repair pathways could be responsible for indel formation and correction from oligo templates, and we therefore rephrased the text in the revised manuscript accordingly. In the revised manuscript we also used NGS to analyse the repair outcomes in a more direct and precise manner. Of note, our main goal in this study is the enhance precise repair from donor oligos (potentially for future clinical application of CRISPR/Cas), and we are aware that we cannot draw conclusions about the activity of specific molecular repair pathways. Since ss-oligos are more efficient for HDR correction of DSBs than ds-DNA, we also solely focused on ss donor oligos in our study.

It is claimed that NHEJ is error-prone. This is wrong. There are two kinds of end-joining processes, the canonical NHEJ (which is not error-prone, but conservative) and the Alternative end-joining (alt-NHEJ, MMEJ, B-NHEJ), which is mutagenic and error-prone. Therefore mutagenic repair does not automatically imply NHEJ.

We fully agree, and want to thank the reviewer for pointing out this issue. In the revised manuscript we rephrased the text accordingly (see Introduction).

Second it is said in the Introduction that HR and NHEJ directly compete. This is also wrong; in fact things happen in two phases: first competition between cNHEJ and single-strand DNA resection, second on resected DNA extremities, competition between HR and alternative end-joining. Finally, HR can also generate mutagenesis. Therefore many concerns exist on the strategy used here (which is the sole assay used) because mutagenic repair can arise by many other processes than NHEJ.

We fully agree with reviewer 3; it is not a direct competition between HDR and NHEJ pathways. We were referring to an indirect effect: Indel mutations disable the CRISPR-Cas9 complex from re-targeting the same locus, and thus precise repair from the donor oligo is no longer possible. We rephrased the text in the revised manuscript.

Moreover, is correction with oligonucleotides (65, 81 b) an actual HR mechanism? This is not consistent with concept of MEPS.

Again we agree with reviewer 3. In the revised manuscript we define homology-directed repair (HDR) as any repair pathway that uses a donor oligo as a template for the repair. Interestingly a recent study suggests that ss-donor oligos are predominantly used by the Fanconi Anemia (FA) repair pathway (doi: https://doi.org/10.1101/136028) rather than the classical homologous recombination (HR) pathway.

The authors should first genetically validate their reporter system, in cells mutated for HR or NHEJ.

We initially validated the reporter system by quantifying the increase in correction efficiencies between the classical Cas9 system and the Cas9-Geminin system. Cas9-Geminin expression is limited to the S/G2 phase of the cell cycle, and therefore HDR rates are enhanced (doi: 10.1016/j.celrep.2016.01.019). Importantly, when using plasmid donors as well as ss-oligo donors we could reproduce these results (see Author response image 3), validating that our reporter is able to pick up increases in correction rates.

**Author response image 3. respfig3:** 

The authors should also verify their strategy with natural endogenous target sequences, instead of the reporter.

In the revised manuscript we targeted 5 endogenous loci (Figure 5A-C, Figure 6D=F) (see also our second response to major points).

There are no data on the transfection efficiency. Especially, comparing the CAS9 with the engineered one.Similarly, does the two CAS9 cleave with similar efficiency?

Images of an experiment where a *Sp*Cas9-mCherry fusion protein was transfected using lipofectamine 2000 is shown in Author response image 4 (the upper panel). In the lower panel, we included the comparison of the cleavage efficiencies between classical *Sp*Cas9 WT and the engineered Cas9-SNAP. Furthermore, in the revised version of the manuscript the comparison can be seen in Figure 6C and Figure 4—figure supplement 2B.

**Author response image 4. respfig4:** 

Does the modification of the CAS9 affect its cleavage specificity?Are there any off-target effects? (off-target cleavage, off-target integration of the oligo).

This is indeed a very important question. In the revised manuscript we compared DNA editing rates at the top 3 predicted off-target loci between the classical Cas9 WT (with the uncoupled oligo) and our engineered Cas9-SNAP with the coupled oligo (Figure 6C). Importantly, no increase in off-target editing was observed.

The authors should test different RNA guides for a common target.

We have initially tested 7 different guide RNAs that bind in the mutated RFP locus (using the system from Mashiko et al., 2013) and from there we pre-selected 2 guides that were the most efficient (and were binding over the mutRFP fluorophore). Next, we tested those 2 guides for cleavage in the stably integrated reporter system (see FACS plots in Author response image 5) and selected sgRNA*^Sp^*^Cas9^(mutRFP)-1, which was 1.3 fold more efficient for all further experiments. Additionally, sgRNA*^Sp^*^Cas9^(mutRFP)-1 was the only sgRNA that allowed introduction of the DSB directly in the fluorophore of mutated RFP. This is important for two reasons: (1) positioning the DSB away from the targeted nucleotide (fluorophore of mutated RFP) would reduce the correction efficiencies; (2) introducing the mutation in the protospacer sequence of the guide prevents the guide from reintroducing the DSB.

**Author response image 5. respfig5:** 

The author should test different cell lines.

We fully agree with reviewer 3 that these experiments would allow to validate the robustness or our system. As stated above, in the revised manuscript we analysed three different cell types (HEK cells, K562 cells, mouse ES cells) (Figure 5A-C, Figure 6D-F).

It is not clear what is actually measured. Where are the mutagenic repair (pseudo-NHEJ) measurements? Is it the frequency of HDR or the ratio HDR/pseudo-NHEJ?

We thank the reviewer for pointing that the definition was not clear enough. In the revised manuscript we included the more detailed explanation (Materials and methods; FACS analysis). The correction efficiency (percentage of corrections in edited cells) = 100 * (turboRFP positive population / (eGFP negative population + turboRFP positive population)).

[Editors’ note: the author responses to the re-review follow.]

The manuscript has been improved but there are some remaining issues that need to be addressed before acceptance, as outlined below:The authors provide convincing answers to all reviewers' comments. Importantly, the authors show that the main improvement achieved by their experimental approach is due to enhanced targeting of the oligonucleotides to the nucleus rather that its targeting to the specific target genomic site. This could also be highlighted in the Abstract.

In the revised version of the manuscript we have now added the information that increased correction is due to targeting the repair template to the nucleus in the Abstract.

A few points need to be clarified:1) Source data 6 refers to Figure 4—figure supplement 2 (not 3?) should be registered as Figure 4—source data 3.2) Source data 5 refers to Figure 4—figure supplement 1 (not 2?) should be registered as Figure 4—source data 2.

Thank you for suggesting these clarifications. We have now renamed the figure supplements and source data files in order to be consistent.

3) Figure 2B requires clarification (and in the legend as well). Legend says "the mutation substitutes… CT.. to TA". This is ambiguous because TA is wild type. It would clarify to draw in Figure 2, the wild type and mutated genomic sequences (with the codon substitution) and the sequence of the guide (also can be confusing as drawn because the guide has CT not TA).

In the revised manuscript we now illustrated the RFP reporter sequence before and after the correction (see Figure 2A, B). In addition we also marked the nucleotides and amino acids that are exchanged upon repair.

4) Figure 5: In Figure 5, only the reference is shown. Mutated variant should be indicated. For instance, on 5A, at HBB, the DNA sequence shown should be the one after correction (to be consistent with Figure 2B) ? However it is identical to guide sequence?

Thank you for this important suggestion. As for the reporter system, we have now drawn the schemes for all other endogenous sequences before and after the correction (see Figure 5A, B, C and 6D, E, F).

Please explain the correction efficiency values, obtained after correction for transfection efficiency? If so, provide transfection efficiencies, otherwise read data from Supplementary file 2 cannot be understood: i.e. 27% at Rosa26, how was this obtained?

Thank you for spotting that we forgot to precisely explain how we calculated the correction efficiency values from NGS data. We have now included the formula for this calculation in the Materials and methods section: Correction efficiency (%) = 100* Perfectly corrected/Edited reads.

Using this formula and the values from the Supplementary file 4 (categorized variant count tables), one can calculate the correction efficiencies at the Rosa26 locus: 11.78%, 10.16%, and 7.8% for the uncoupled control and 19.29%, 20.51% and 15.86% for our coupled system. This on average gives a correction efficiency of 9.9% and 18.6%, respectively (presented in the Figure 5B and Figure 5—source data 1).

Of note, the correction efficiency represents the ratio between precise correction and total edits, and therefore does not change with different transfection rates (if we would correct for transfection efficiency in the formula, we would correct the denominator and the numerator by the same value, and thus the result would remain the same).

In addition, since no HR reads is detected at Rosa26 (why is this?), the authors should comment in the main text on the difference uncoupled/coupled from Figure 5B for this locus.

We are very thankful for spotting this mistake in the allele plot for the Rosa26 locus; the corrected sequence (SNV:-2C,-1T) was by mistake plotted close to the bottom of the graph. In the revised version we placed the corrected allele in the 3^rd^ row, as described in the figure legend and as shown in the other plots.

By having a close look at the count table for the Rosa26 locus, we also realized that we made an error when categorizing alleles in the count table (some indels were counted as reference – Supplementary file 4). In the revised version we fixed this error, and adapted the values accordingly. Importantly, these changes did not substantially affect our results, and did not change any of the conclusions of the manuscript.

5) Figure 6A requires clarifications both in the main text and in the presentation of the figure. The authors summarize in one sentence all the NGS data (Results and Discussion, eighth paragraph). There is a lot of data in this analysis and it would be better to present the analysis step by step and not to refer to all panels 6A, B, D, E, F at once. Reporter locus could be presented first, with comments about percentage of corrected reads and other events (legend should explain the nomenclature, for instance -7:7D, and also what are I, II and III: triplicates I assume). Then results at other loci should be briefly discussed (and referred to Supplementary file 2).

Thank you very much for this important comment. As suggested, in the revised manuscript we discuss the data shown in Figure 6 in more detail in the main text (Results and Discussion, eighth paragraph), and refer to Supplementary file 2. In addition, in the figure legends we now explain the nomenclature for SNVs and indels, and indicate that the different colors in the x-axis labels represent the three different replicates.

Also indicate in Supplementary file 2 that the variants called "SNV" are the ones predicted after repair by HR.

In the revised version we always specify the precisely corrected variant in the allele plots.

An interesting information is also the relative proportion of HR versus non HR events. In Figure 6, it seems that this proportion is about 30% in experiment I (5.63/(100-85.19)). Is this correct? It could be discussed in the main text.

In the revised version we explain the editing results of the experiment shown in Figure 6A in more detail: We found that while the mean percentage of corrected loci increased from 0.8% with the classical Cas9 system to 4.9% with the RNPD system, the number of incorrectly edited loci slightly decreased from 12.6% to 9.3%, respectively (Figure 6A, Supplementary files 2, 3, 4). This corresponds to a 7-fold increase in correction efficiency from 5.3% to 36.4% (see Figure 6B).